# ATP sensing in living plant cells reveals tissue gradients and stress dynamics of energy physiology

Valentina De Col[1,2], Philippe Fuchs[1], Thomas Nietzel[1], Marlene Elsässer[1], Chia Pao Voon[3], Alessia Candeo[4], Ingo Seeliger[1], Mark D Fricker[5], Christopher Grefen[6], Ian Max Møller[7], Andrea Bassi[4], Boon Leong Lim[3,8], Marco Zancani[2], Andreas J Meyer[1,9], Alex Costa[10], Stephan Wagner[1]*, Markus Schwarzländer[1,9]*

[1]Institute of Crop Science and Resource Conservation, University of Bonn, Bonn, Germany; [2]Department of Agricultural, Food, Environmental and Animal Sciences, University of Udine, Udine, Italy; [3]School of Biological Sciences, University of Hong Kong, Hong Kong, China; [4]Dipartimento di Fisica, Politecnico di Milano, Milano, Italy; [5]Department of Plant Sciences, University of Oxford, Oxford, United Kingdom; [6]Centre for Plant Molecular Biology, Developmental Genetics, University of Tübingen, Tübingen, Germany; [7]Department of Molecular Biology and Genetics, Aarhus University, Aarhus, Denmark; [8]State Key Laboratory of Agrobiotechnology, Chinese University of Hong Kong, Hong Kong, China; [9]Bioeconomy Science Center, Forschungszentrum Jülich, Jülich, Germany; [10]Dipartimento di Bioscienze, Università degli Studi di Milano, Milano, Italy

*For correspondence: stephan.
wagner@uni-bonn.de (SW);
markus.schwarzlander@uni-bonn.
de (MS)

Competing interests: The
authors declare that no
competing interests exist.

Reviewing editor: Christian S.
Hardtke, University of Lausanne,
Switzerland

**Abstract** Growth and development of plants is ultimately driven by light energy captured through photosynthesis. ATP acts as universal cellular energy cofactor fuelling all life processes, including gene expression, metabolism, and transport. Despite a mechanistic understanding of ATP biochemistry, ATP dynamics in the living plant have been largely elusive. Here, we establish MgATP$^{2-}$ measurement in living plants using the fluorescent protein biosensor ATeam1.03-nD/nA. We generate Arabidopsis sensor lines and investigate the sensor in vitro under conditions appropriate for the plant cytosol. We establish an assay for ATP fluxes in isolated mitochondria, and demonstrate that the sensor responds rapidly and reliably to MgATP$^{2-}$ changes in planta. A MgATP$^{2-}$ map of the Arabidopsis seedling highlights different MgATP$^{2-}$ concentrations between tissues and within individual cell types, such as root hairs. Progression of hypoxia reveals substantial plasticity of ATP homeostasis in seedlings, demonstrating that ATP dynamics can be monitored in the living plant.

## Introduction

ATP is universal in cells. It is used as a metabolic building block and as a cofactor to couple exergonic and endergonic reactions, making ATP as fundamental to life as proton gradients and the genetic code. An additional function as a biological hydrotrope to keep proteins soluble was further suggested (*Mandl et al., 1952*; *Patel et al., 2017*). De-phosphorylation to ADP and AMP, and re-phosphorylation to ATP allow high energy fluxes based on relatively small pool sizes in the cell (*Rich, 2003*). The major sites of ATP synthesis are ATP synthases driven by the proton motive force established across the inner membrane of the mitochondria, and the plastid thylakoids in plants. The

ATP produced typically fuels the wide range of energy-demanding processes, such as motility, transport, and gene expression, in other parts of the cell, although plastids also consume substantial amounts of ATP in the Calvin-Benson cycle, to an extent that varies substantially with the light/dark cycle. Coupling of these interactions requires ATP/ADP exchange between the organelles and cytosol. Thus, regulation of cytosolic and organelle ATP levels, and ATP/ADP transport across the mitochondrial and plastid envelope, give rise to particularly complex ATP dynamics in plant cells, that is critically dependent on the tissue type and external conditions (*Neuhaus et al., 1997*; *Flügge, 1998*; *Reiser et al., 2004*; *Haferkamp et al., 2011*). The exact nature of the interplay between mitochondria and chloroplasts in maintaining cytosolic and nuclear ATP homeostasis, especially under changeable conditions, such as light–dark cycles or varying $O_2/CO_2$ status, has been investigated for decades, predominantly using biochemical techniques or in vivo NMR (*Bailleul et al., 2015*; *Gardeström and Igamberdiev, 2016*). For example, insight into subcellular adenine nucleotide pools have been possible through rapid membrane filter-based fractionation of leaf protoplasts, revealing a complex and dynamic interplay between the three cell compartments (*Lilley et al., 1982*; *Stitt et al., 1982*; *Gardeström and Wigge, 1988*; *Krömer and Heldt, 1991*; *Krömer et al., 1993*). The current consensus is that cytosolic ATP is mainly provided by the mitochondria, both in the dark, but also in the light, when photorespiration can be the main driver of ATP synthesis (*Igamberdiev et al., 2001*). Little is known, however, about differences between organs, tissues and cells and about the characteristics of their specific responses over time.

Similar challenges apply to other physiological and metabolic parameters, such as pH, free $Ca^{2+}$, potentials of thiol redox couples, and concentrations of small molecules including plant growth regulators. However, development of in situ reporters has provided increasingly sophisticated understanding of their in vivo behaviour (*De Michele et al., 2014*; *Uslu and Grossmann, 2016*). For example, detailed insights into subcellular pH gradients and their dynamics have turned out to play a critical role in membrane transport, protein degradation, and energy and ion homeostasis (*Schwarzländer et al., 2012*; *Luo et al., 2015*). Likewise, the spatiotemporal characteristics of free $Ca^{2+}$ transients are central to signalling in stress responses and plant-microbe interactions (*Choi et al., 2014b*; *Keinath et al., 2015*). The ability to separately monitor redox potentials of the subcellular glutathione pools has revealed a far more reducing cytosolic redox landscape than previously anticipated, and has led to novel concepts of redox regulation and signalling (*Marty et al., 2009*; *Morgan et al., 2013*; *Schwarzländer et al., 2016*).

Recently, several different fluorescent sensor proteins for ATP have been engineered (*Berg et al., 2009*; *Imamura et al., 2009*; *Kotera et al., 2010*; *Nakano et al., 2011*; *Tantama et al., 2013*; *Yoshida et al., 2016*). 'Perceval' is based on a single circularly permuted mVenus protein fused to the bacterial regulatory protein GlnK1 from *Methanococcus jannaschii*. Competitive binding of ATP and ADP to GlnK1 result in inverse changes of two excitation maxima to provide a ratiometric read-out of ATP:ADP (*Berg et al., 2009*). The 'PercevalHR' variant was obtained by mutagenesis, and has an improved dynamic range of about 4 (*Tantama et al., 2013*). However, both variants are strongly pH-sensitive, requiring pH measurement and correction for meaningful in vivo measurements. By contrast, the ratiometric ATeam sensor family was introduced as far less pH-sensitive. ATeam sensors share their overall design with the widely used Förster Resonance Energy Transfer (FRET) sensors of the Cameleon family, making use of the ε-subunit fragment of ATP synthase from *Bacillus* sp. PS3 for reversible ATP binding (*Imamura et al., 2009*; *Kotera et al., 2010*). ATP binding to the ε-subunit induces a conformational change in the sensor structure modifying the relative orientation of the N- and C-terminal donor and acceptor fluorophores (monomeric super-enhanced cyan fluorescent protein (mseCFP); circularly permuted monomeric Venus (cp173-mVenus), a variant of yellow fluorescent protein), increasing FRET efficiency. Both sensor classes have provided insights into subcellular ATP dynamics in animals (*Ando et al., 2012*; *Tarasov et al., 2012*; *Li et al., 2015*; *Merrins et al., 2016*). However, to date, there is only one report on their use in plants (*Hatsugai et al., 2012*), and establishment of reliable fluorescence-based ATP monitoring in plants has been lacking. This is despite the prominent role of ATP in the physiological network of plants, including two ATP-producing organelles and frequently fluctuating environmental conditions that determine the development of their flexible body plan.

In this work, we set out to establish protocols for ATP sensing in plants. First, we generate Arabidopsis lines expressing the biosensor ATeam1.03-nD/nA in the cytosol, mitochondrial matrix or the plastid stroma, and demonstrate that plants harbouring the probe in the cytosol or plastids are

stable and show no phenotypic change. By contrast, lines expressing mitochondrial sensors are dwarfed, but still viable. Second, we validate the biochemical characteristics of the ATeam 1.03-nD/nA sensor in vitro under conditions typically encountered in plant systems. Third, we develop an ex situ assay for isolated mitochondria to probe ATP transport and synthesis. Fourth, we map tissue differences and gradients of cytosolic $MgATP^{2-}$ concentrations in living seedlings, including cell-to-cell variation in root hairs that inversely correlates with the rate of growth. Finally, we demonstrate how progressive hypoxia leads to characteristic time-dependent changes in $MgATP^{2-}$ dynamics.

## Results

### Generation of Arabidopsis lines for ATP sensing in the cytosol, chloroplasts and mitochondria

To establish ATP measurements in living plants, we generated Arabidopsis lines expressing ATeam1.03-nD/nA in the cytosol, the chloroplast stroma, and the mitochondrial matrix. We selected at least three independent lines for each compartment based on fluorophore expression, two of which were propagated to homozygosity. Despite expression being driven by a CaMV 35S promoter, we did not observe strong sensor silencing in subsequent generations contrary to frequent observations for other sensors (*Pei et al., 2000*; *Deuschle et al., 2006*; *Chaudhuri et al., 2008*; *Yang et al., 2010*; *Jones et al., 2014*; *Behera et al., 2015*; *Loro et al., 2016*; *Schwarzländer et al., 2016*). Fluorescence in the peripheral cytoplasm, as well as in trans-vacuolar strands and the nucleoplasm, demonstrated cytosolic expression, whilst co-localisation with chlorophyll auto-fluorescence confirmed chloroplast expression (*Figure 1A*). Furthermore, all independent lines for cytosol or plastid expression showed a wild-type-like phenotype at the whole plant level (*Figure 1B*), which was validated by detailed phenotyping quantifying root length, rosette size, inflorescence height and number of siliques (*Figure 1—figure supplements 2* and *3*). By contrast, transformants for mitochondrial expression showed a consistently weaker fluorescence and a strong developmental phenotype (*Figure 1A,B*; *Figure 1—figure supplement 2*). The sensor fluorescence co-localised with the mitochondrial matrix marker MitoTracker, but the organisation of the labelled organelles was perturbed, suggesting abnormal some degree of mitochondrial abnormality. Nevertheless, transformants flowered after 14 weeks and set seed, allowing their propagation. Despite these observations, we recorded the fluorescence of Venus and CFP in five-day-old seedlings (*Figure 1—figure supplement 1*) and found markedly lower Venus/CFP ratios in mitochondria while cytosol and chloroplasts were similar (*Figure 1C*; *Figure 1—figure supplement 1*).

### In vitro characterisation of purified sensor protein revealing specificity for $MgATP^{2-}$

To interpret in vivo measurements in the Arabidopsis sensor lines, we aimed for an in-depth understanding of key sensor characteristics. The $K_d(ATP)$, nucleotide specificity and pH sensitivity of the original ATeam family variants were characterised for use in animal cells at 37°C (*Imamura et al., 2009*). While the newer ATeam1.03-nD/nA variant shows an improved dynamic range (*Kotera et al., 2010*), its other properties have not been characterised in detail, particularly at the pH and temperature conditions likely to be experienced in the plant cytosol, mitochondrial matrix or plastid stroma. We therefore characterised purified ATeam1.03-nD/nA protein in vitro (*Figure 2A*). The sensor emission spectrum showed a well-defined ratiometric shift in response to ATP. The mseCFP peak (475 nm) decreased and the cp173-mVenus peak (527 nm) increased, with increasing ATP concentrations in the high micromolar/low millimolar range (*Figure 2B*). The isosbestic point was at 512 nm. Purified protein was stable at −86°C and retained the same dynamic range of the freshly purified sensor. However, non-frozen storage caused a decline of sensor responsiveness over time, arising from sensor degradation, probably by proteolysis, explaining the diminished dynamic range by separation of the two chromophores (*Figure 2—figure supplement 1A*). Hence, aliquots of purified protein were frozen immediately after purification and stored at −86°C. The ATP response of the sensor, determined at 25°C, was sigmoidal with a spectroscopic dynamic range of 4.0, higher than previously reported at 3.2 (*Kotera et al., 2010*). The $K_d(ATP)$ was 0.74 mM with a Hill coefficient of 1.02, compatible with a single ATP binding site in the ATP synthase ε-subunit (*Yagi et al., 2007*) (*Figure 2C*). The sensor showed no response to ADP and AMP, in agreement with previous reports using other

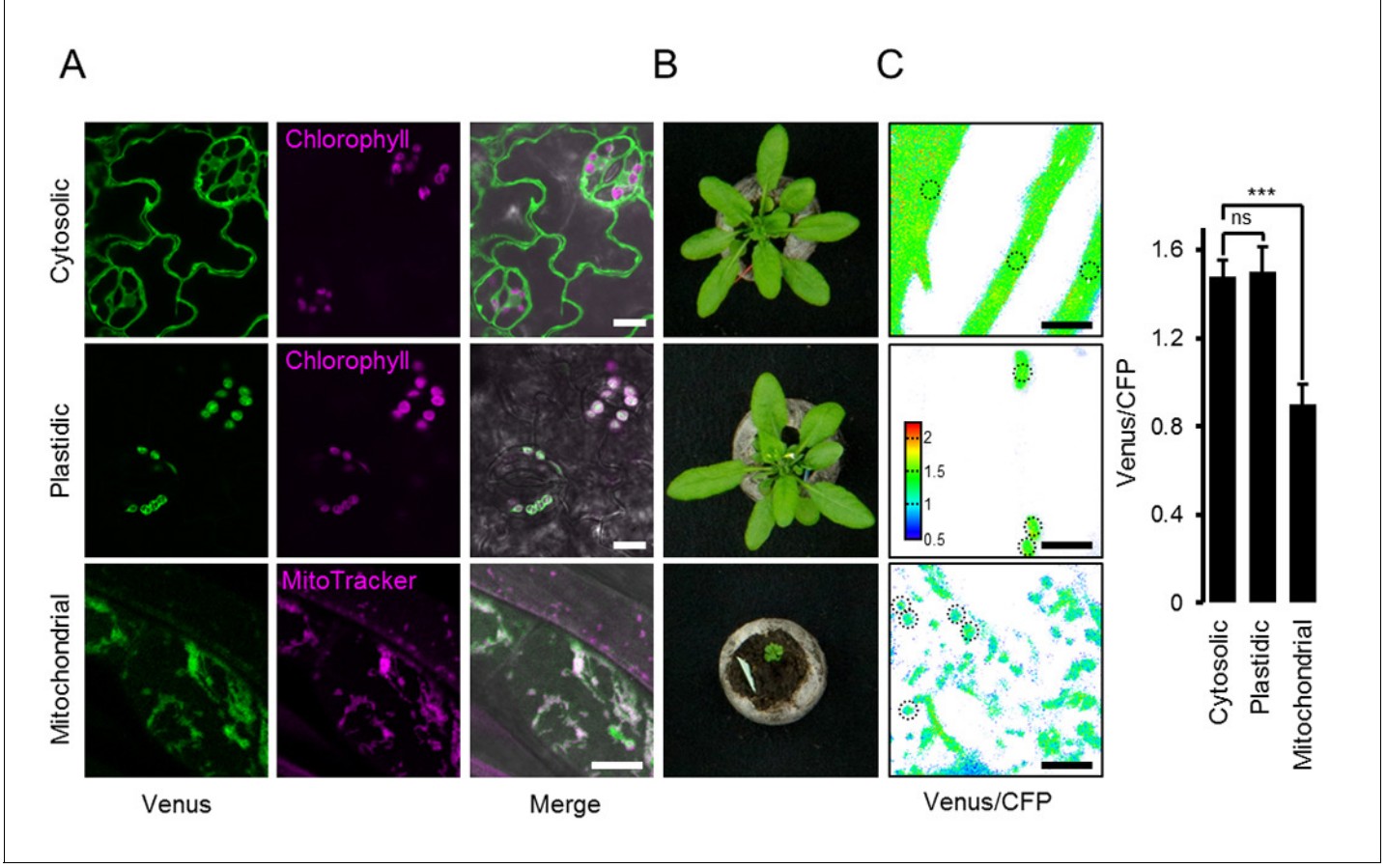

**Figure 1.** ATeam expression in stable Arabidopsis lines. ATeam1.03-nD/nA was expressed under the control of a 35S promoter as unfused protein for localisation in the cytosol, as a fusion with the transketolase target peptide (TkTp) for plastid targeting or fused to the *Nicotiana plumbaginifolia* β-ATPase for mitochondrial targeting. (**A**) Five-day-old seedlings grown vertically on half-strength MS +1% (w/v) sucrose medium plates were used for CLSM. Venus (green) fluorescence was recorded alongside the chlorophyll fluorescence in cotelydon cells or the mitochondrial marker MitoTracker Orange in cells of the hypocotyl. The merge image shows both fluorescence channels projected on the respective bright field image. (**B**) Phenotypes of multiple independent lines per construct were compared and a representative image is shown after growth for five weeks on soil. (**C**) For a ratiometric analysis, fluorescence of Venus and CFP was assessed in hypocotyl cells of five-day-old seedlings grown as in (**A**) with power of the 458 nm laser set to 10% (cytosolic and plastidic) and 30% (mitochondrial) of maximal power. Regions of interest (ROIs) of similar size, indicated by dotted lines, were defined to calculate the Venus/CFP ratio shown in the graph. $n$ = 36 (cytosol/plastid) or 105 (mitochondria) ROIs in 12 (cytosol/plastid) or 22 (mitochondria) images from 4 (cytosol/plastid) or 6 (mitochondria) individual plants; error bars = SD. ns: $p > 0.05$, ***$p \leq 0.001$ ($t$ test). Scale bar (all panels) = 10 µm.

The following figure supplements are available for figure 1:

**Figure supplement 1.** Ratiometric imaging of ATeam in cellular compartments of Arabidopsis seedlings.

**Figure supplement 2.** Whole plant phenotyping of homozygous ATeam lines.

**Figure supplement 3.** Whole plant phenotyping of heterozygous ATeam lines.

**Figure supplement 4.** Phenotyping of Arabidopsis expressing mitochondrial sensor proteins.

nucleotides, including GTP (*Imamura et al., 2009*), and confirms the selectivity for ATP. $Mg^{2+}$ titration under saturating ATP showed a strong $Mg^{2+}$ dependence, indicating that the sensor responds selectively to $MgATP^{2-}$, and not to $ATP^{4-}$ (*Figure 2D,E*). In the absence of ATP, the sensor signals were pH-stable from pH 6.5 to 8.5 (*Figure 2F*), consistent with the absence of direct pH effects on the fluorophores. Insensitivity to pH was also observed at partially saturating and saturating ATP

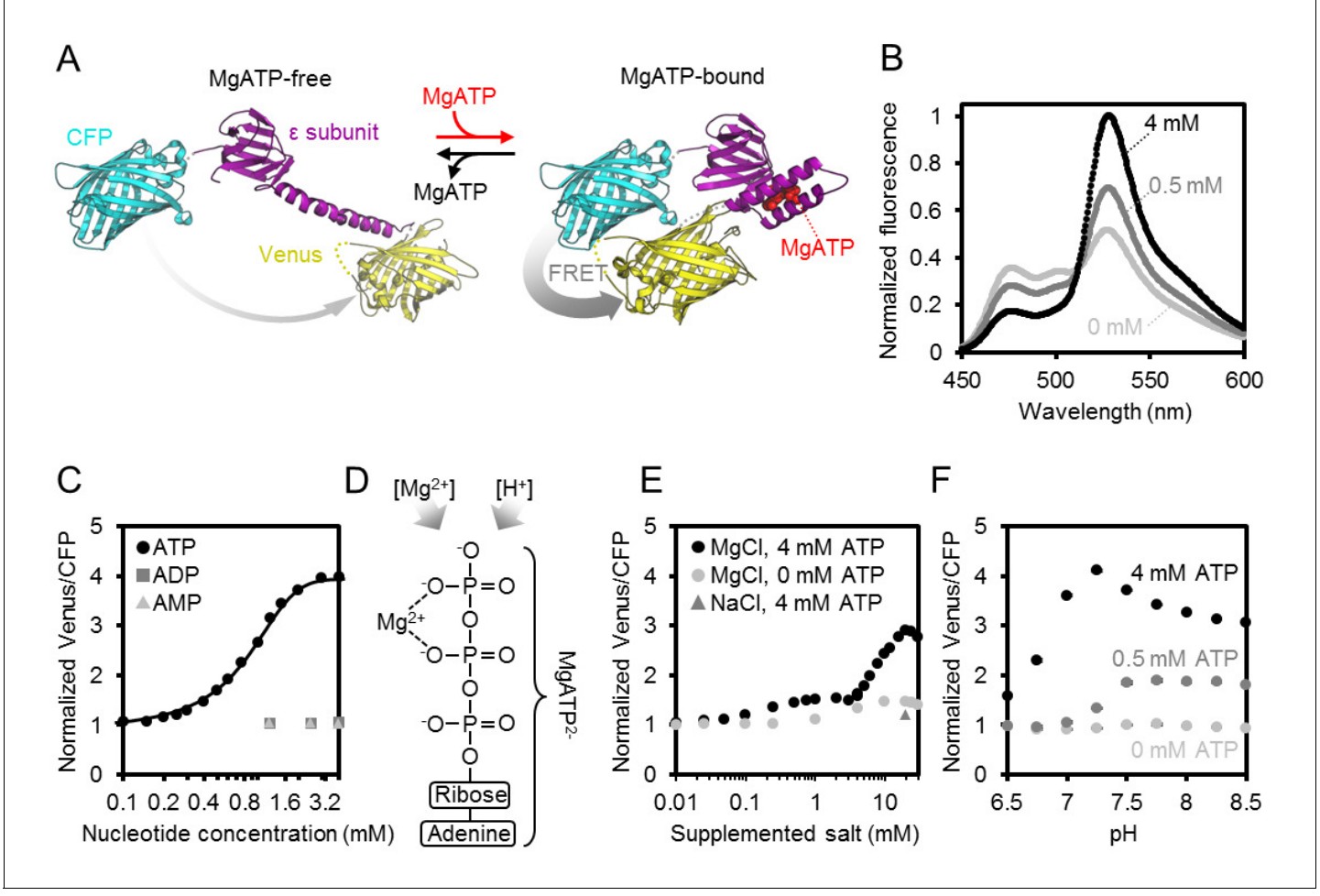

**Figure 2.** Characteristics of purified ATeam1.03-nD/nA. (**A**) A cyan fluorescent protein (CFP, PDB: 2WSN) and a variant of the yellow fluorescent protein (Venus, PDB: 3EKJ) were manually linked by the ε-subunit of *Bacillus subtilis* ATP synthase in the $MgATP^{2-}$-bound (PDB: 2E5Y) and $MgATP^{2-}$-free (PDB: 4XD7) state to generate a hypothetical structural model of ATeam. FRET efficiency in the absence or presence of $MgATP^{2-}$ is indicated by a grey arrow. (**B**) Normalised ATeam emission spectra (excitation at 435 ± 5 nm) in the presence of increasing ATP concentrations and an excess in $Mg^{2+}$ by 2 mM. (**C**) ATeam was excited at 435 ± 5 nm and the ratio of emission at 527 nm (cp173-Venus) and 475 nm (mseCFP) at 25°C in the presence of adenine nucleotides is plotted. The Boltzmann function was used to fit $MgATP^{2-}$-binding data. (**D**) Structure of $MgATP^{2-}$. Its stability depends on pH and on the free $Mg^{2+}$ concentration. (**E**) ATeam Venus/CFP ratios in 4 mM ATP (black points) and 0 mM ATP (grey points) titrated with increasing concentrations of $MgCl_2$. The grey triangle shows the Venus/CFP ratio in the presence of 4 mM ATP and 20 mM NaCl. (**F**) ATeam Venus/CFP ratios at different pH and in the presence of 0 (light grey), 0.5 (dark grey) and 4 mM (black) MgATP. Data in (**B**), (**C**) and (**F**) is averaged from four technical replicates and error bars are represented as SD, but too small to be displayed.

The following figure supplement is available for figure 2:

**Figure supplement 1.** Characteristics of purified ATeam1.03-nD/nA.

from pH 7.5 to 8.5, although the spectroscopic sensor response range was diminished below pH 7.0. This decrease in ratio was due to a bona fide FRET response (mseCFP donor signal increasing, cp173-mVenus acceptor signal decreasing; *Figure 2—figure supplement 1B*), and correlated with the decrease in the $MgATP^{2-}$ species as the pH was lowered. We infer that the sensor response reports the $MgATP^{2-}$ concentration, which itself depends on ambient $Mg^{2+}$ concentration and pH (*O'Sullivan and Perrin, 1961*; *Storer and Cornish-Bowden, 1976*; *Adolfsen and Moudrianakis, 1978*). Since $MgATP^{2-}$ is the species that acts as the cofactor for the large majority of ATP-dependent proteins, the sensor provides a readout of the physiologically significant proportion of the ATP

pool that is visible for those proteins. The *MgATP²⁻ response* of the sensor will be referred to as *ATP response* for simplicity in sections of this work.

## Establishing an ex situ assay to monitor mitochondrial ATP dynamics

In vivo ATP dynamics in the cell are governed by production and consumption. On the production side, mitochondria export ATP to the cytosol (and the nucleus in turn by diffusion), facilitated by oxidative phosphorylation at the matrix surface of the inner mitochondrial membrane and membrane gradient-driven ATP extrusion by a very active ADP/ATP carrier (AAC) system (*Haferkamp et al., 2011*; *Gout et al., 2014*). We hypothesised that supplementing purified, functional mitochondria with external sensor protein would allow monitoring of ATP transport fluxes into and out of the mitochondria, and also dissect the role of adenylate kinase (AK), which is thought to be localised to the intermembrane space (*Figure 3A*). Such a system would remove the influence of other cellular processes that impact on MgATP²⁻ concentration in the cytosol, including ATP hydrolysis, transport across membranes of other cell compartments, as well as changes in pH and Mg²⁺ concentrations.

We supplemented freshly isolated Arabidopsis seedling mitochondria with purified recombinant ATeam sensor protein using the same medium as used for in vitro sensor characterisation (*Figure 3B*). The medium contained Mg²⁺ and P$_i$ in excess, and was set to pH 7.5 to

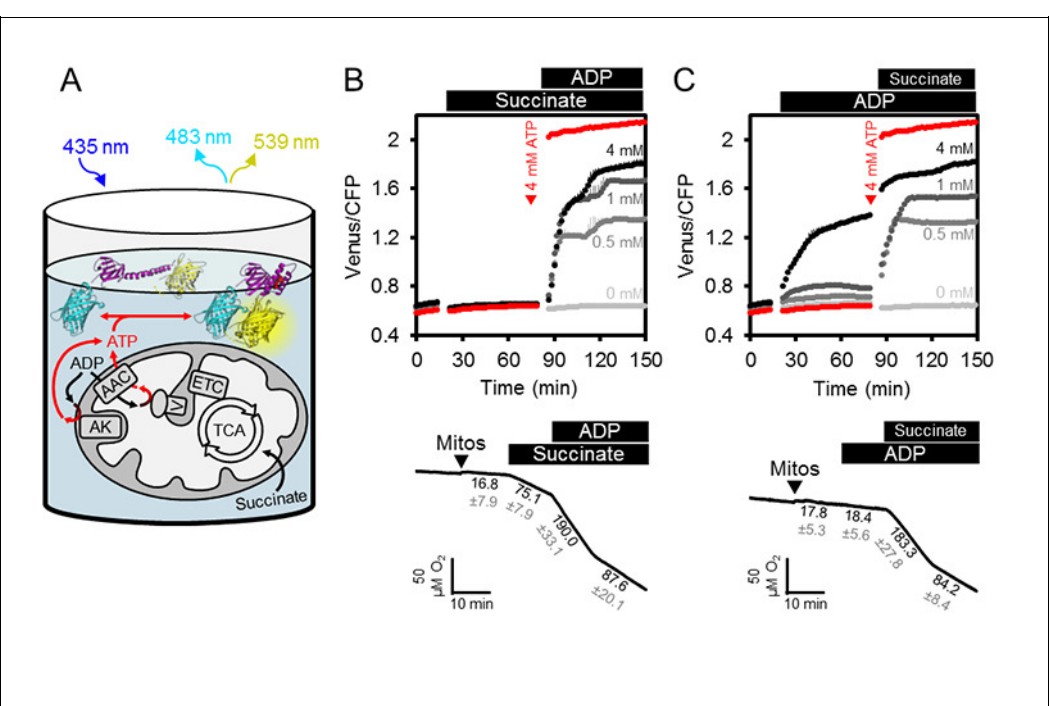

**Figure 3.** ATP fluxes in isolated Arabidopsis mitochondria. (A) Pure intact mitochondria isolated from two-week-old Arabidopsis seedlings were mixed with basic incubation medium and purified ATeam in 96-well microtiter plates. Mitochondria were fed with succinate as a respiratory substrate to fuel the tricarboxylic acid (TCA) cycle and the electron transport chain (ETC). ATP, chemiosmotically generated by ATP synthase (complex V) in the matrix, is exchanged for ADP through the ADP/ATP carrier (AAC), while ATP generated by adenylate kinase (AK) in the intermembrane space does not require transport across the inner mitochondrial membrane. Extramitochondrial ATP exported from the mitochondria is sensed by ATeam in the medium as MgATP²⁻. (B, C) ATeam was used as shown in (A) and 10 mM succinate was added either before (B) or after (C) ADP at concentrations between 0 and 4 mM. ATP was added to 4 mM as reference at the indicated time point (red traces). n (technical replicates) = 4; error bars = SD. Lower panels show polarographic oxygen consumption assays performed with a Clark-type electrode in parallel. Mitochondria, succinate and ADP were added to the basic incubation medium as indicated. A representative trace from an individual experiment is shown and oxygen consumption rates (nmol min⁻¹ mg⁻¹ protein) for each respiratory state are given as mean ± SD from three technical replicates.

resemble cytosolic pH (*Ratcliffe, 1997*; *Schulte et al., 2006*). The exact concentration of the sensor is not critical, because the ratiometric FRET readout is self-normalising, although very low or very high sensor concentrations were avoided, to prevent low signal-to-noise, or ATP-buffering by the sensor itself when close to its $K_d$(ATP).

The FRET ratio did not change on addition of succinate as respiratory substrate in the absence of ADP, but subsequent addition of ADP led to an increase in FRET that plateaued at a steady state value depending on the ADP concentration (*Figure 3B*). Polarographic oxygen consumption assays performed in parallel confirmed respiratory activity responses and coupling of the mitochondria with a respiratory control coefficient of around 2 (*Figure 3B*), and confirmed that the sensor responded rapidly to ATP generated by respiring mitochondria and exchanged by the AAC. Nevertheless, addition of ADP before succinate revealed that ADP alone was sufficient to cause the sensor to respond in a dose-dependent manner, although with smaller FRET increases (*Figure 3C*). Considering the sensor does not respond to ADP (see *Figure 2C*), the response is indicative of ATP production in the absence of active respiration. FRET ratios increased further after subsequent addition of succinate, reaching a similar plateau value to before. Production of ATP from ADP alone in the absence of a respiratory substrate ruled out ATP synthase activity, but would be consistent with AK activity leading to conversion of ADP to ATP and AMP (*Busch and Ninnemann, 1996*). This implies that the assay provides an integrated readout of the combined activities of ATP synthase, AK and the AAC.

## Dissection of mitochondrial ATP production, ATP/ADP exchange, and adenylate kinase activity

The mitochondrial assay allows investigation of the role of AK and AAC in controlling mitochondrial ATP dynamics. To test whether AK is involved in ATP production in the absence of respiratory electron transport, we predicted that ATP production would be sensitive to the presence of AMP driven by mass action, in a fully reversible reaction catalysed by AK (*Figure 4A*). Consistent with this view, addition of AMP following the increase in FRET ratio triggered by ADP, led to a gradual, dose-dependent decrease of FRET (*Figure 4B*). Likewise, the presence of AMP before addition of ADP, inhibited ADP-induced ATP generation (*Figure 4C*).

To test the role of AAC-mediated ADP/ATP exchange across the inner membrane (*Figure 4D*), we added carboxyatractyloside (cAT) to block the AAC (*Figure 4E*). Addition of 4 mM ADP in the absence of respiratory activity resulted in a similar FRET increase in both control and cAT-treated mitochondria, suggesting that AK-derived ATP did not rely on AAC-mediated ADP or ATP transport. Subsequent energization by succinate lead to a slight but reproducible further increase in the control, which was absent for the cAT-treated mitochondria. This difference shows that matrix-exposed ATP synthase cannot contribute in the presence of cAT and validates effective AAC inhibition. At lower concentrations of ADP (0.5 mM), when ATP production by AK is close to the detection limit (see *Figure 3C*), succinate gave an increase in detectable MgATP$^{2-}$ in the absence of cAT, but this was almost completely abolished in the presence of cAT (*Figure 4E*). This is consistent with localisation of AK outside the inner mitochondrial membrane and supports localisation in the mitochondrial intermembrane space, as reported previously for other plant species (*Day et al., 1979*; *Birkenhead et al., 1982*; *Stitt et al., 1982*; *Roberts et al., 1997*; *Zancani et al., 2001*), but contrasts the situation in mammals where AK3 isoforms are also present in the matrix (*Schulz, 1987*).

By exploiting the differential sensitivity of AK and the AAC/ATP synthase system to ADP, it is possible to minimise the contribution of AK to ATP production, and use low ADP concentrations (0.5 mM) to selectively monitor ATP produced by ATP synthases and AAC activity (*Figure 4C*). ATP synthesis then strictly depends on ADP import, ATP synthase functionality, the proton motive force and functional electron transport (*Figure 4F*). Treatment with specific inhibitors that cover those four functional levels consistently prevented the FRET increase in response to mitochondrial energization by succinate (*Figure 4D–F*), validating the assay as a means to monitor functional changes at specific steps in bioenergetic pathways.

## High-throughput measurements of MgATP$^{2-}$ in planta by fluorimetry

To allow for high-throughput measurements of MgATP$^{2-}$ levels in planta, we optimised a microtiter plate fluorimetry setup for plant tissues and whole seedlings. Ratiometric analysis makes measurements independent of sensor expression level, as well as tissue amount and shape, provided there is

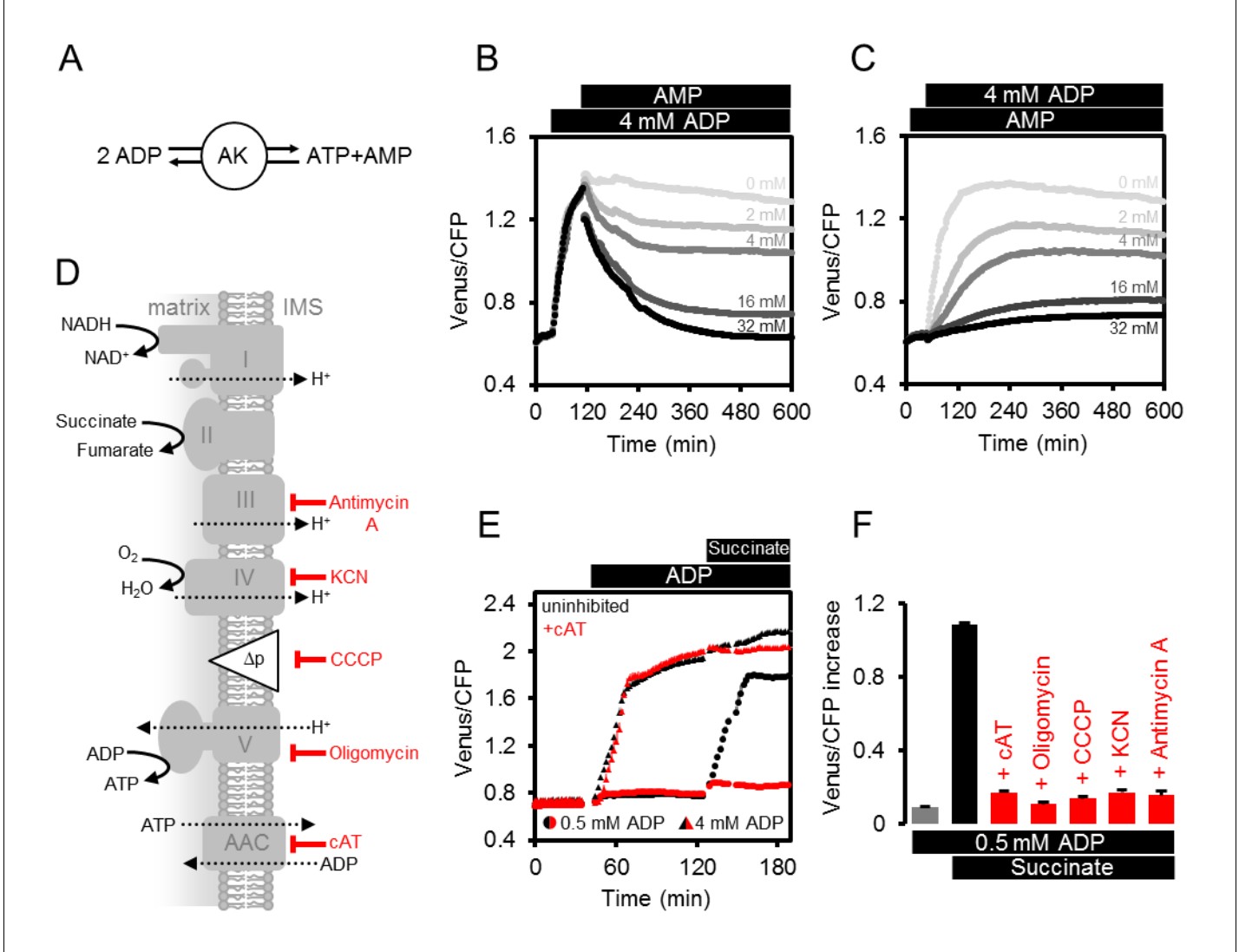

**Figure 4.** Modulating the ATP production in isolated Arabidopsis mitochondria. (A) Reaction catalysed by adenylate kinase (AK). (B, C) ATeam was used in a setup as shown in *Figure 3A*. Isolated mitochondria and purified ATeam were mixed with AMP between 0 (light grey) and 32 (black) mM either after (B) or before (C) the addition of 4 mM ADP and the ATeam Venus/CFP ratio was recorded. *n* (technical replicates) = 4. (D) Representation of the mitochondrial electron transport chain (complexes I-IV) that generates a proton motive force ($\Delta$p) used by the ATP synthase (complex V) to produce ATP. The ADP/ATP carrier (AAC) transports ATP from the mitochondrial matrix to the intermembrane space (IMS) in exchange for ADP. Treatments that diminish mitochondrial ATP production or transport and their site of action are indicated. (E) Untreated mitochondria (black symbols) or mitochondria treated with 10 μM cAT (red symbols) were fed with ADP at 0.5 mM (circles) or 4 mM (triangles) followed by 10 mM succinate. (F) The inhibitory effect of treatments summarised in (F) on mitochondrial ATP production was calculated through the Venus/CFP increase after addition of ADP (grey), ADP and succinate (black) or ADP and succinate under inhibition (red). cAT, carboxyatractyloside; CCCP, carbonyl cyanide m-chlorophenyl hydrazone; KCN, potassium cyanide. E and F show the mean of three technical replicates; error bars = SD.

sufficient signal-to-noise and little interference from tissue auto-fluorescence. Emission spectra were recorded with excitation at 435 nm from seven-day-old intact Arabidopsis seedlings (*Sweetlove et al., 2007*), and leaf disks of four-week-old plants expressing cytosolic ATeam (*Figure 5A,B*). Both sample types showed fluorescence spectra that were practically identical with the purified sensor protein, while auto-fluorescence was low by comparison. Chlorophyll fluorescence was effectively separated and did not cause any significant interference. This was independently confirmed also for the chloroplastic sensor by confocal microscopy at the individual chloroplast level (*Figure 5—figure supplement 2*).

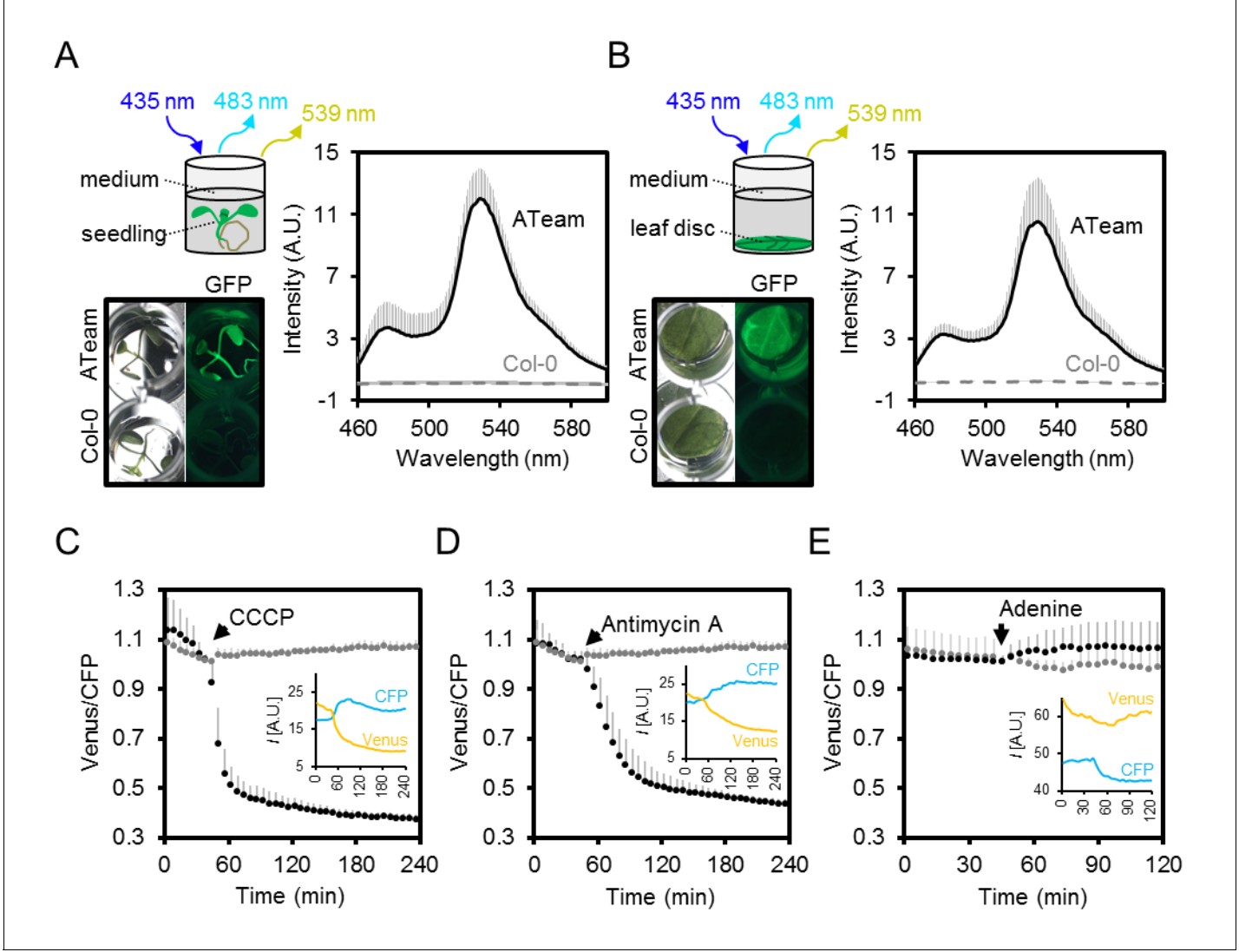

**Figure 5.** Fluorimetry setup to monitor MgATP$^{2-}$ dynamics in planta. (**A,B**) Seven-day-old Arabidopsis seedlings (two per well; **A**) or leaf discs of individual four-week-old plants (**B**) were submerged in imaging medium on 96-well microtiter plates. Fluorescence of plant material stably expressing cytosolic ATeam was checked with an epifluorescence microscope equipped with a GFP filter and compared to wild-type Col-0. Fluorescence emission spectra between 460 and 600 nm were recorded using a plate reader and an excitation wavelength of 435 ± 10 nm. $n$ = 5; error bars = SD. (**C–E**) Per well, two seven-day-old Arabidopsis seedlings expressing no sensor (Col-0) or cytosolic ATeam were excited at 435 ± 10 nm and the emission at 483 ± 9 nm (mseCFP) and 539 ± 6.5 nm (cp173-Venus) was recorded. CCCP (100 μM), antimycin A (100 μM) or adenine (10 mM) were added where indicated (black data points) while control plants were left untreated (grey data points). Emission in wells with Col-0 plants was averaged and subtracted from that of ATeam-expressing plants to correct for background fluorescence. Data shown and used for background subtraction is the average of 3–4 wells and error bars are SD. Insets show the fluorescence emission intensity ($I$) of Venus and CFP in representative individual wells.

The following figure supplements are available for figure 5:

**Figure supplement 1.** Fluorometric readings of MgATP$^{2-}$ dynamics in Arabidopsis seedlings.

**Figure supplement 2.** Impact of chlorophyll fluorescence on the ATeam signal.

**Figure supplement 3.** The effect of CCCP and antimycin A treatments on the ATeam response at different medium pH.

To assess the speed and range of the sensor response in vivo, we monitored the FRET ratio of seedlings in microtiter plates over time in the presence and absence of carbonyl cyanide m-chloro-phenyl hydrazone (CCCP), antimycin A, and adenine, to modify endogenous $MgATP^{2-}$ concentrations. CCCP dissipates proton gradients over cell membranes, inhibiting ATP production and increasing ATP consumption; antimycin A inhibits mitochondrial electron transport at complex III (*Figure 4D*); while adenine feeding has been demonstrated to lead to an increase of the cellular $MgATP^{2-}$ concentration by acting as a substrate for ATP synthesis (*Loef et al., 2001*; *Gout et al., 2014*). We focused on the cytosol as an integration space for ATP fluxes from and to other subcellular locations, where steady-state concentration of $MgATP^{2-}$ is set by the interplay of numerous synthesis, hydrolysis and transport processes.

CCCP and antimycin A both triggered a rapid and pronounced decrease in FRET, while adenine led to a modest, but reproducible increase (*Figure 5C,D,E*; *Figure 5—figure supplement 1*). Varying the medium pH between 6.0 and 8.5 showed that the decrease in FRET after antimycin A treatment was independent of pH. Also after CCCP addition, only a small fraction of the response was due to cytosolic acidification, and destabilisation of $MgATP^{2-}$ in turn (*Figure 5—figure supplement 3*). The maximal spectroscopic response range was about 3, slightly lower than the range in vitro, which is partially accounted for by the excitation wavelength and emission bandwidth used. We infer that the sensor was intact and functional in vivo. High, but not fully saturated FRET, at steady state allowed an estimate of cytosolic $MgATP^{2-}$ concentrations in the range of about 2 mM, averaged over seedling tissues. This is consistent with previous estimations and textbook values (*Taiz et al., 2015*), but higher than reported in other cases (*Gout et al., 2014*).

## A $MgATP^{2-}$ map of the Arabidopsis seedling

Heterogeneity and gradients between tissues and cells have been of major interest in plant hormone signalling and development, and fluorescent sensors for abscisic acid and auxin were introduced recently (*Brunoud et al., 2012*; *Wend et al., 2013*; *Jones et al., 2014*; *Waadt et al., 2014*). Analogous insights are largely lacking for metabolites and co-factors, despite the fact that metabolism underpins development (*Sweetlove et al., 2017*), and that abiotic factors, such as hypoxia, act as signals (*Considine et al., 2017*). To measure potential differences in cytosolic $MgATP^{2-}$ concentrations between tissues and cells in vivo, we performed a confocal microscopy analysis of intact five-day-old Arabidopsis seedlings (*Figure 6A*). The seedlings were kept in the dark for 30 min before image acquisition to avoid potential effects of active photosynthesis. An overview of FRET across the seedlings revealed large tissue differences. Cotyledons displayed high values, which were lower in the hypocotyl and dropped abruptly at the shoot–root transition. The root showed low values which then increased in the root tip. The relative differences were reproducibly observed between individual seedlings, and consistent in independent sensor lines (#1.1, #3.6; *Figure 6B*; *Figure 6—figure supplement 1*). The area around the shoot apex showed low ratios, comparable with those of the root, in many but not all individuals. By contrast, differences between tissues were less pronounced in etiolated five-day-old seedlings under identical conditions (*Figure 6C,D*), with comparatively lower FRET in the shoots and higher FRET in the roots. The steep gradient at the shoot–root transition was also absent, indicating that photo-morphogenesis is required for its formation.

To test whether the ratios observed could be attributed to $MgATP^{2-}$ concentrations rather than optical artefacts from the different tissue geometries, or other biochemical modifications of the sensors, CCCP was used to deplete $MgATP^{2-}$. In addition, the medium buffer pH was set to 7.5 to avoid destabilisation of $MgATP^{2-}$ as a result of cytosolic acidification. In the presence of CCCP, FRET ratios decreased towards a similar minimum value in all tissues and tissue heterogeneity was gradually abolished (*Figure 6—figure supplement 2A,B*). Tissue-specific kinetics that could be resolved by CLSM may reflect differential capacity of tissues to maintain $MgATP^{2-}$ concentrations or simply differential penetration by CCCP. To independently validate the overall difference between shoot and root, we used the microtiter well fluorimetry approach to measure shoot and root samples separately, as attempted previously in tissue extracts (*Mustroph et al., 2006*). The full spectra indicated bona fide FRET differences, which were abolished by CCCP treatment (*Figure 6—figure supplement 2C,D*). It is important to note that relative and normalised FRET changes, but not absolute FRET values, can be compared between different fluorimetry/microscopy setups due to different excitation and emission detector configurations.

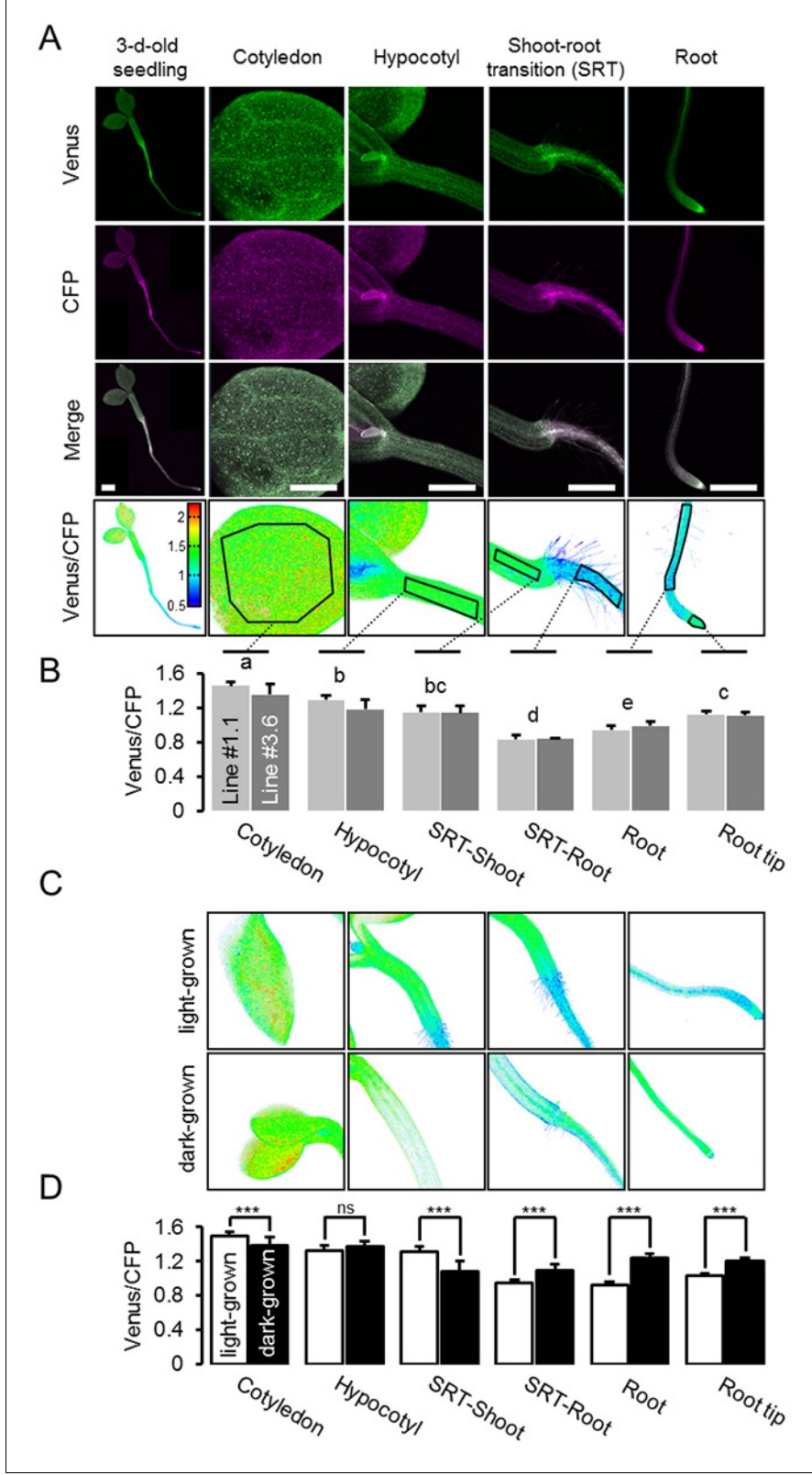

**Figure 6.** A MgATP²⁻ map of the Arabidopsis seedling. (**A**) To map the MgATP²⁻ levels in tissues of Arabidopsis seedlings, three-day-old (whole seedling) or five-day-old (close-ups) plants expressing cytosolic ATeam were analysed by CLSM. Fluorescence of Venus (green) and CFP (magenta) was recorded and the ratio is plotted as a false-color image where high Venus/CFP values (red) correspond to high MgATP²⁻ levels. In the close-ups, Venus/
*Figure 6 continued on next page*

*Figure 6 continued*
CFP ratios were analysed in the indicated regions of interest. Scale bar = 500 µm. (B) Graphs represent data from two independent lines. *n* (per line) = 6; error bars = SD. Data from both lines were pooled for a statistical analysis with a one-way ANOVA followed by the Tukey test (p≤0.05) and different letters indicate significant differences. The experiment was repeated two times with consistent results. (C, D) CLSM analysis of five-day-old seedlings either grown in the light or etiolated in the dark. SRT, shoot-root transition. *n* (seedlings per condition) = 11; error bars = SD. ns: p>0.05, ***p≤0.001 (two-way ANOVA followed by the Tukey test).
The following figure supplements are available for figure 6:

**Figure supplement 1.** A MgATP$^{2-}$ map of the Arabidopsis seedling.
**Figure supplement 2.** The effect of CCCP on the ATeam response in different Arabidopsis seedling tissues.

The conclusion that in vivo FRET heterogeneity reliably registered MgATP$^{2-}$ heterogeneity, prompted us to further resolve differences at the cellular level. Comparing pavement and guard cells of abaxial cotyledon epidermis by a region of interest (ROI) analysis did not reveal any differences in our hands (*Figure 7A*). A previous report from older, true leaves had shown higher MgATP$^{2-}$ concentrations in guard cells than in pavement cells, suggesting that differences can be induced depending on developmental and environmental conditions (*Hatsugai et al., 2012*). Analysis of seven cell layers of the shoot–root transition zone revealed a continuous gradient, indicating that neighbouring cells can maintain different, but stable, MgATP$^{2-}$ gradients (*Figure 7B*). At the root tip, cap cells showed low MgATP$^{2-}$. Similarly, low FRET values were observed at wounding sites (*Figure 7C*). A hotspot of high MgATP$^{2-}$ levels was localised in the columella just below the quiescent centre (*Figure 7D*).

## Root hair growth speed is correlated with MgATP$^{2-}$ levels as indicated by light sheet fluorescence microscopy

High respiration has been found during pollen tube growth (*Dickinson, 1965*) and the same is likely for other rapidly tip-growing cells, such as root hairs. On the one hand, high respiration rates may give rise to high ATP levels; on the other, high demand during rapid growth may nevertheless cause ATP depletion. Addressing the question of the relationship between ATP levels and growth of single cells has previously been technically impossible. We employed light sheet fluorescence microscopy (LSFM) for 4D imaging of the growing Arabidopsis root to quantify growth rates and FRET of the individual root hairs (*Figure 8A*). We measured both growth speed and FRET ratio of individual hair cells (*Figure 8B*). While the most rapidly growing 20% of hair cells elongated eight times more quickly than the slowest 20%, the latter showed an increase in their FRET ratio indicating that their cytosolic MgATP$^{2-}$ concentration was increased (*Figure 8C,D*). Indeed, the average growth speed was inversely correlated with cytosolic MgATP$^{2-}$ content and lowered MgATP$^{2-}$ was exclusively detected in rapidly growing hair cells (*Figure 8E*). Interestingly, there was no evidence for oscillations in MgATP$^{2-}$ levels similar to those observed for Ca$^{2+}$ (*Hepler et al., 2001*; *Monshausen et al., 2008*; *Candeo et al., 2017*), as FRET ratios remained stable and did not show any periodic oscillations in the order of seconds, as validated by Fourier analysis (*Figure 9*).

## In vivo monitoring of the decline in cytosolic MgATP$^{2-}$ levels during hypoxia

Several stress conditions have been correlated with a cellular energy crisis. Hypoxia has a direct impact on mitochondrial ATP production, since lack of oxygen as final electron acceptor inhibits the respiratory chain. However, flexible metabolic responses have been described in response to hypoxia to prolong maintenance of cellular energy supply (*Geigenberger et al., 2000*; *Geigenberger, 2003*; *van Dongen et al., 2009*; *Zabalza et al., 2009*; *van Dongen and Licausi, 2015*). To monitor MgATP$^{2-}$ dynamics during the progression of hypoxia, we used oxygen-proof, transparent tape to seal medium-filled wells containing individual seedlings grown on agar plates and in liquid culture, leaving no residual air space (*Figure 10*; *Figure 10—figure supplement 1*). The experiments were carried out in the dark (except for excitation flashes for fluorimetric readings) to avoid oxygen evolution by photosynthesis. Seedlings in non-sealed wells served as controls. All FRET ratios

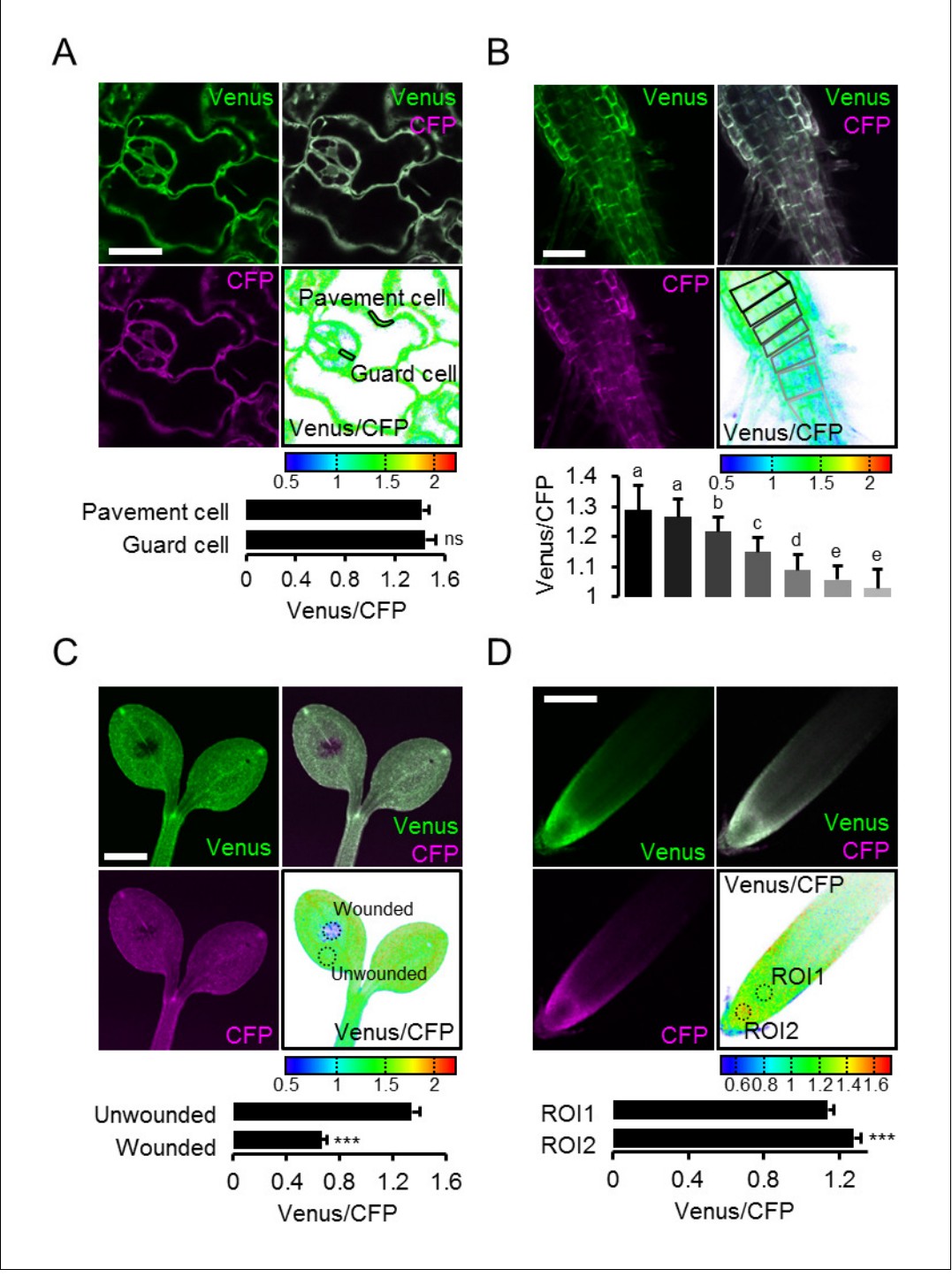

**Figure 7.** Local MgATP$^{2-}$ heterogeneity in Arabidopsis seedling cells and tissues. Fluorescence of Venus (green) and CFP (magenta) from cytosolic ATeam was recorded by CLSM in five-day-old Arabidopsis seedlings. The ratio is plotted as a false-color image where high Venus/CFP values (red) correspond to high MgATP$^{2-}$ levels. (**A**) Venus/CFP ratios were assessed in indicated regions of interest to compare guard cells and pavement cells of the cotyledon epidermis. $n$ = 24 pairs of pavement and guard cell from five individual plants; error bars = SD. ns: p>0.05 (*t* test). Scale bar = 20 µm. (**B**) Venus/CFP ratios were assessed in indicated cell layers at the shoot–root transition. Region of interest analysis of successive cell layers is indicated in grey scale. $n$ = 11; error bars = SD. Different letters indicate statistical differences in a one-way ANOVA followed by the Tukey test (p≤0.05). Scale bar = 100 µm. (**C**) Cotyledons were wounded with a needle and Venus/CFP ratios were assessed in the indicated

*Figure 7 continued on next page*

*Figure 7 continued*

regions of interest representing wounded or unwounded tissue. *n* = 5; error bars = SD, ***p≤0.001 (*t* test). Scale bar = 500 μm. (**D**) The Venus/CFP ratio was assessed in two regions of interest at the root tip. *n* = 12; error bars = SD, ***p≤0.001 (*t* test). Scale bar = 100 μm.

showed an initial decrease immediately after immersion. This was followed by a phase of steady decrease for sealed seedlings, while unsealed seedlings retained FRET ratios around the starting values. Re-oxygenation by removal of the seal allowed full recovery to control values, while maintaining the seal led to a phase of sharp decrease and plateauing of FRET ratios at low values, similar to

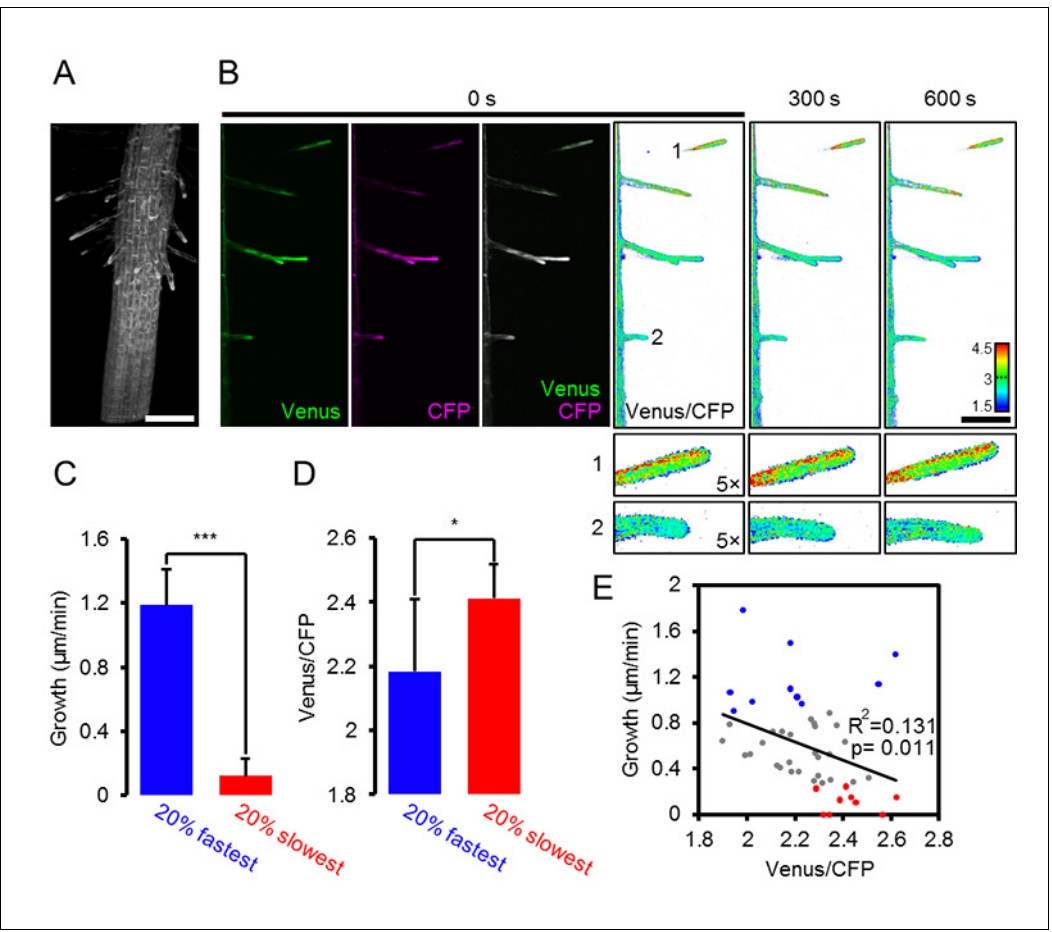

**Figure 8.** Light sheet fluorescence microscopy (LSFM) analysis of MgATP$^{2-}$ levels in growing root hair cells. Hair cell growth on roots from six-day-old seedlings expressing cytosolic ATeam was followed for 10 min by LSFM. (**A**) Maximum projection of the Venus signal in the elongation and lower maturation zone generated from 4D in vivo imaging data of the growing Arabidopsis root. Scale bar = 100 μm. (**B**) Representative root area with hair cells in different developmental stages. Fluorescence of Venus (green) and CFP (magenta) was recorded and the ratio is plotted as false-color images over three time points. Scale bars = 100 μm. Five times magnified image sections exemplify (1) a slow-growing and (2) a fast-growing hair cell. (**C**) Hair cells in the top and bottom 20% quantile interval of growth speed and (**D**) the corresponding cytosolic ATeam FRET ratio measured in the tip of the same cells. *n* = 48 hair cells from six roots; error bars = SD, *p≤0.05, ***p≤0.001 (*t* test). (**E**) Growth speed of individual hair cells as function of their cytosolic ATeam FRET ratio. Values of the slowest and fastest growing cells as included in (**C**) and (**D**) are indicated in red and blue; line indicates linear regression. Significance of correlation, expressed as a p-value determined by an *F* test, and the coefficient of determination (R$^2$) based on all data points are provided.

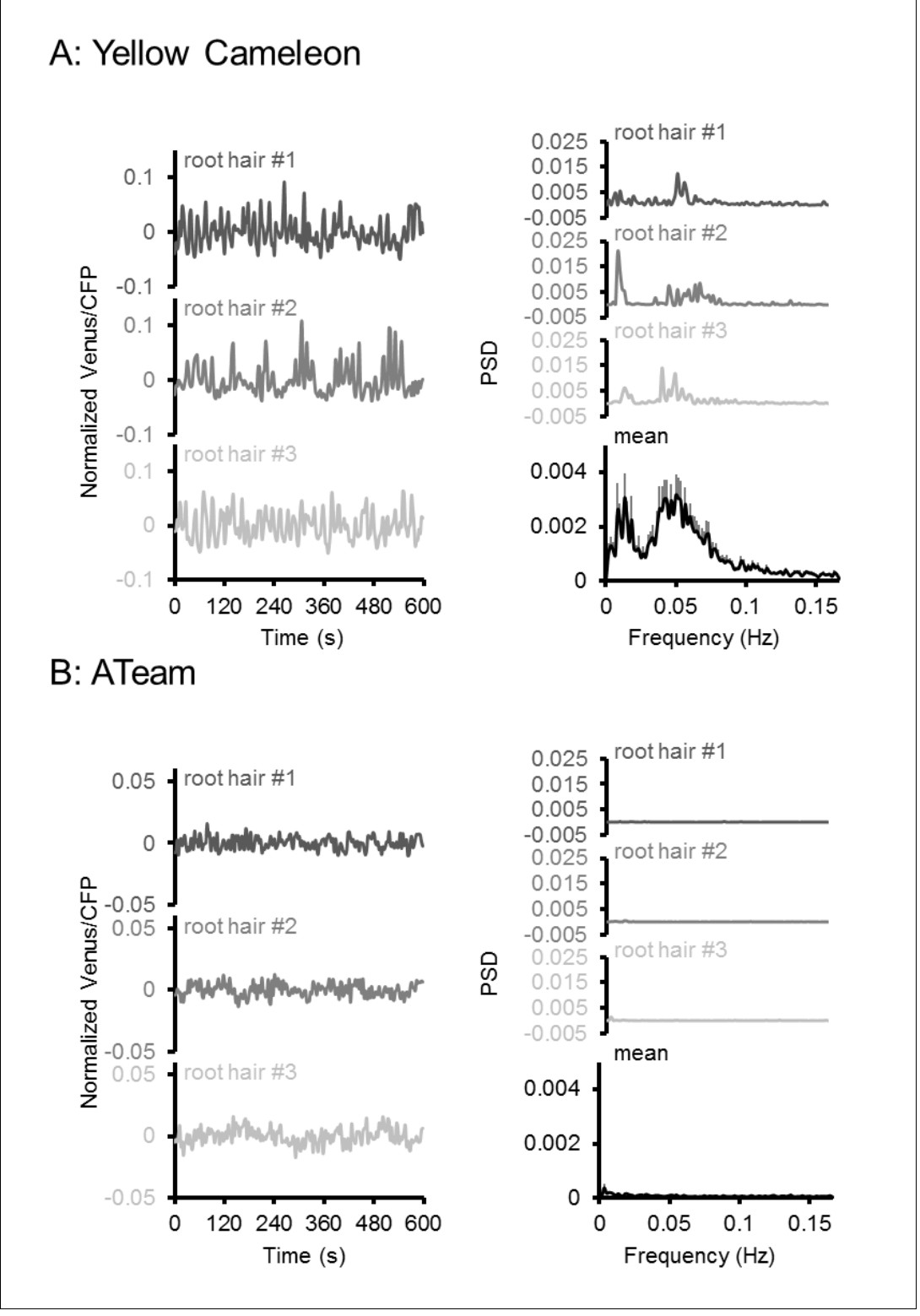

**Figure 9.** Oscillation analysis in root hair cells. FRET analyses in root hair tips of six-day-old seedlings expressing (**A**) NES-Yellow Cameleon 3.6 and (**B**) cytosolic ATeam1.03-nD/nA by LSFM. Oscillations of three representative root hairs are shown as normalised Venus/CFP ratios and power spectral density (PSD) spectra by Fourier analysis. The mean PSD spectra represent 22 root hairs each. Error bars = SEM. Note the presence of typical $Ca^{2+}$ oscillations around 0.05 Hz in the Cameleon dataset, which were absent in the ATeam dataset.

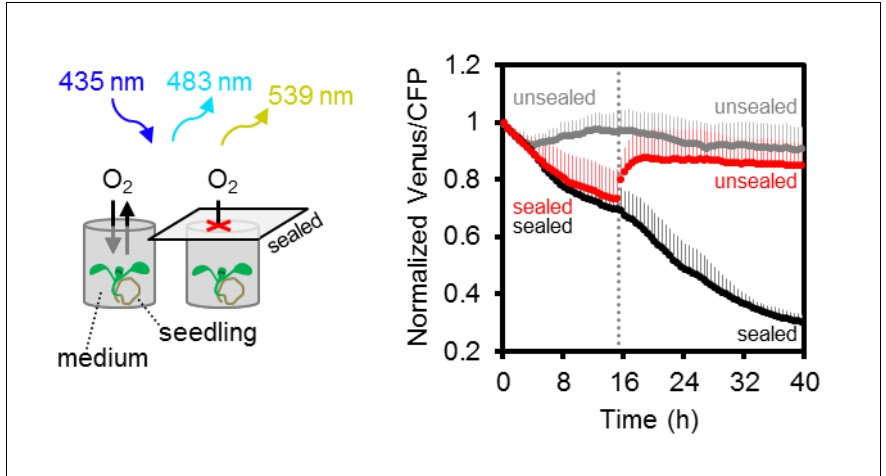

**Figure 10.** Arabidopsis MgATP[2-] dynamics under low oxygen. Ten-day-old Arabidopsis seedlings, grown vertically on plates, were submerged in imaging medium on 96-well microtiter plates. Per well, three seedlings expressing no sensor (Col-0) or cytosolic ATeam were excited at 435 ± 10 nm, and the emission at 483 ± 9 nm (mseCFP) and 539 ± 6.5 nm (cp173-Venus) was recorded. Wells were either left open (grey), sealed with an oxygen-proof, transparent qPCR film (black) or sealed for 15.5 hr before the film was removed to reoxygenate the samples (red). Emission in wells with Col-0 plants was averaged and subtracted from that of ATeam-expressing plants to correct for background fluorescence. Data shown and used for background subtraction is the mean of 9–12 wells and error bars are SD.

The following figure supplement is available for figure 10:

**Figure supplement 1.** Arabidopsis MgATP[2-] dynamics under low oxygen.

---

those seen for CCCP treatment. While the response characteristics were reproducible, the onset and rates of the different phases varied between individual seedlings, probably as a result of differences in biomass, respiration rate and the rate of oxygen decrease in turn. The hypoxia model demonstrates the possibility to reliably monitor subcellular MgATP[2-] dynamics live during stress insult, and may be flexibly adjusted to other external conditions, tissues and genetic backgrounds.

## Discussion

### Fluorescent monitoring to understand ATP dynamics

Fluorescent ATP sensors were initially engineered for mammalian cells and tissues (*Berg et al., 2009*; *Imamura et al., 2009*; *Kotera et al., 2010*; *Nakano et al., 2011*; *Tantama et al., 2013*), where their use has allowed novel insights, such as subcellular ATP concentration gradients between compartments in situ, visualisation of stimulus-induced energy dynamics in neurons, responsiveness of ATP-sensitive K[+] channel activity in single cells and synchronisation of Ca[2+] and ATP dynamics in HeLa cells with histamine stimulation. With the exception of one report on cell death induced by hypersensitive response (*Hatsugai et al., 2012*), fluorescence ATP measurements have been lacking in plant research. Live-cell fluorescent monitoring complements widespread standard approaches, such as luminescent or HPLC-based ATP determinations in biological extracts, or radioisotope-based techniques for membrane transport assays (*Manfredi et al., 2002*; *Khlyntseva et al., 2009*; *Lorenz et al., 2015*; *Monné et al., 2015*). While those techniques offer high sensitivity and accuracy in quantifying ATP, and other nucleotides, in extracts, they provide endpoint measurements after removal from the functional biological system and have limited use for resolving the ATP concentration over time at the (sub)cellular and tissue level. Yet, high flux rates, rapid fluctuations and (sub)cellular gradients are fundamental characteristics of cellular energy physiology. Non-destructive live measurement of ATP has been available by [31]P-NMR spectroscopy, which offers the additional benefit of also measuring ADP and other di- and tri-nucleotides (*Gout et al., 2014*). However, [31]P-NMR is

limited in spatial resolution and sensitivity. The superior sensitivity of fluorescence-based MgATP$^{2-}$ monitoring is demonstrated in the experiments with isolated Arabidopsis mitochondria (*Figure 3*), where changes in the MgATP$^{2-}$ concentration were monitored with time resolution in the order of seconds on only 20 µg of mitochondrial protein. The spatial resolution of fluorescent ATP sensing is highlighted by its ability to detect MgATP$^{2-}$ levels in individual cells (*Figure 7*) and single chloroplasts (*Figure 1*; *Figure 5—figure supplement 2*), making ATP-binding probes a particularly versatile technique in tissue and cell physiology.

## Arabidopsis lines for monitoring MgATP$^{2-}$ in the cytosol, chloroplasts and mitochondria

Governance of cytosolic ATP levels by two bioenergetic organelles in green plant cells raises important questions about the regulatory basis of subcellular ATP control (*Gardeström and Igamberdiev, 2016*). The cytosolic sensor lines indicated no differences in ratio between cytosol and nucleoplasm, indicative of rapid diffusion between the two locations and in line with previous observations for free Ca$^{2+}$ concentrations and glutathione redox potential (*Loro et al., 2012*; *Schwarzländer et al., 2016*). The chloroplast sensor lines provide a good FRET signal, even in the presence of chlorophyll (*Figure 5—figure supplement 2*). These lines will be a valuable tool to dissect the impact of photosynthesis on subcellular ATP, and investigations are currently underway in our labs. While no developmental phenotype was observed for the cytosolic and plastidic sensor lines (*Figure 1*; *Figure 1—figure supplements 2* and *3*), the strong phenotype that systematically resulted from sensor targeted to the mitochondrial matrix (*Figure 1C*; *Figure 1—figure supplement 2*) requires caution. It cannot be ruled out that ATP homeostasis is generally perturbed in those sensor lines, although sensor-based measurements in mammalian cells have also indicated lowered ATP in the matrix, which can be accounted for by membrane potential-driven AAC-mediated ATP export (*Imamura et al., 2009*). The reason for the stunted phenotype is currently unclear. Interference with matrix ATP homeostasis by ATP buffering appears unlikely, especially since similar phenotypes of variable severity can also be observed for other mitochondrially targeted sensors (*Figure 1—figure supplement 4*); similar issues were also noted in yeast (*Schwarzländer et al., 2016*). Clogging of the TIM/TOM machinery by import arrest of the synthetic sensor construct may account for the observed effects.

## Sensor characteristics and limitations revealed in vitro

Purified ATeam1.03-nD/nA was selective for MgATP$^{2-}$ (*Figure 2*), refining previous characterisations of the parent sensor ATeam1.03 (*Imamura et al., 2009*). The sensor does not report on energy charge set by the ATP:ADP ratio directly (*Pradet and Raymond, 1983*), without an additional assumption that the total adenosine pool is constant. In vivo, de novo synthesis and degradation are unlikely to change the total adenosine pool as rapidly as ATP/ADP/AMP-cycling by (de-)phosphorylation. Hence, the MgATP$^{2-}$ changes induced by inhibitor treatments and during hypoxia are likely to also reflect decreased energy charge. By contrast, differences in steady-state ATP levels between cells and tissues and organs may reflect constitutive differences in total adenylate pool sizes. Combinatorial analyses with sensors that directly respond to energy charge, such as variants of Perceval (*Tantama et al., 2013*), will be desirable in the future. However, the currently available ATP:ADP sensors suffer from serious pH sensitivity, which adds a level of complexity to the meaningful interpretation in planta, where pronounced pH fluctuations can occur.

## A dynamic assay to dissect ATP fluxes of isolated mitochondria

A key innovation from fluorescent protein-based sensing has been the ability to monitor biochemistry as it occurs in the intact biological system. Nevertheless, mechanistic conclusions have often been problematic based on attempts to link in vitro reductionism with in vivo complexity, and the challenges in bringing intact systems under experimental control. The ATP assays that we establish for isolated mitochondria introduce a useful intermediate (*Figures 3* and *4*). Despite the use of whole organelles as complex multi-functional units, the different steps of ATP dynamics can be monitored and interpreted mechanistically. Substrate feeding and inhibitor treatments allow for tight control over the functional state of the organelle, and controlled manipulation of ATP physiology. For example, we were able to distinguish the phosphorylation activities of ATP synthase and AK,

and monitor either activity by exploiting their differential characteristics, including their sub-mitochondrial exposure to the matrix and the intermembrane space, respectively. Perturbation of mitochondrial function was observed with high sensitivity and kinetically resolved following inhibition of defined players in mitochondrial ATP production. Substrates and inhibitors are well suited to modulate respiratory activity, capacity and efficiency, and to induce transitions in ATP dynamics. They may be applied to dissect bioenergetic rearrangements in respiratory mutants in the future.

The current understanding of cellular bioenergetics has been driven by dynamic measurements in isolated mitochondria and chloroplasts, e.g. of oxygen consumption or $Ca^{2+}$ transport. Yet, monitoring ATP dynamics, as the central product, has typically relied on indirect inference, or reconstruction of time-resolved data from individual samples (*Attucci et al., 1991*). The fluorescent biosensor-based ATP assay concept introduces a complementary method to continuously and dynamically monitor ATP transport fluxes from and into cells and isolated organelles, as demonstrated using plant mitochondria. Since the sensor protein is added externally to the samples, and does not require transgenic sensor expression by the samples, cell components from any species can be studied for ATP generation, consumption and membrane transport. The assays may be further expanded to study membrane transport dynamics. For instance, the response of ATP transport across the chloroplast envelope to changes in photosynthetic activity should be readily detectable making it possible to address as yet unresolved questions about transport efficiency, directionality and routes, under controlled external conditions (*Haferkamp et al., 2011*; *Gardeström and Igamberdiev, 2016*). The assay principle is adaptable to other subcellular structures, and potentially even liposome, cell, and tissue systems. It may also facilitate the improvement of ATP-coupled enzyme assays. Monitoring ATP export and uptake by cells may be of specific interest to address current questions about intercellular ATP fluxes and the signalling function of extracellular ATP (*Kim et al., 2006*; *Tanaka et al., 2010*; *Choi et al., 2014a*). ATeam variants with higher affinities to match lower ATP concentration ranges have been engineered, and are also likely to be specific for $MgATP^{2-}$ (*Imamura et al., 2009*).

## Monitoring ATP in planta by three sensor detection approaches

We have performed live sensing of ATP in planta by three distinct in vivo techniques. Microtiter plate-based fluorimetry (*Figures 3–5* and *10*), CLSM (*Figures 1*, *6* and *7*) and LSFM (*Figures 8* and *9*) complement another to investigate the dynamics and distribution of ATP in living plant tissues. While CLSM has been extensively used in combination with other fluorescent protein sensors in plants, LSFM has only recently been adjusted to enable FRET-based $Ca^{2+}$ measurements in Arabidopsis roots (*Costa et al., 2013*; *Candeo et al., 2017*). Microtiter plate-based fluorimetry has been widely used for cultured animal cells and yeast (*Birk et al., 2013*; *Morgan et al., 2016*), but rarely in plant tissues (*Rosenwasser et al., 2010*, *2011*), and a rigorous technical validation was missing. Our analysis shows that careful adjustment of fluorimetric settings, with the sensor properties and the characteristics of the tissue can make plate reader-based measurements a robust approach for continuous monitoring of fluorescent protein sensors over extended time periods. While heterogeneity between cells and tissues is averaged, large sample numbers can be assessed in parallel enabling robust controls and accounting for biological variability. Genetic targeting of the sensor to a specific subcellular location where it responds to the local ATP levels means that the resolution of subcellular physiology is maintained. Further, side-by-side measurements of different sensors in different tissues or cell compartments may allow monitoring and dissecting the interplay of different physiological responses to the same stimulus. For instance, the impact of hypoxia on cyto-nuclear ATP, pH and $Ca^{2+}$, or the impact of illumination on the ATP concentration in the cytosol and the chloroplast stroma may be assessed. Parallel side-by-side monitoring provides an alternative to multiplexing of several sensors co-expressed in the same plant. The robustness of the assay may even be suitable for genetic or chemical biological screens (*Dejonghe and Russinova, 2017*) based on fluorescent sensors in the future.

## Sensing and manipulating MgATP²⁻ in vivo

Decreasing and increasing the cytosolic $MgATP^{2-}$ levels in vivo by chemical treatments resulted in a reliable response that covered nearly the full theoretical dynamic range of the sensor (*Figure 5*). We infer that the sensor is functional in vivo, and not affected by proteolytic cleavage. The effective

decrease of MgATP$^{2-}$ after inhibition with antimycin A validated the role of mitochondria as major suppliers of cytosolic ATP, as the loss of respiration-derived ATP could not be replaced. Nevertheless, the response kinetics to the chemical treatments appeared to be dominated by tissue uptake rate. Monitoring the treatments by CLSM indicated varying response rates depending on tissue type (*Figure 6—figure supplement 2*). Yet, all tissues gradually adopted a similarly low FRET ratio of the MgATP$^{2-}$-free sensor. The ability to drive the sensor to its extremes in vivo allows the conversion of in vivo FRET ratios to absolute MgATP$^{2-}$ concentrations, as routinely done in Ca$^{2+}$ sensing (*Palmer and Tsien, 2006*), and thiol redox sensing (*Schwarzländer et al., 2008*). Since the conversion requires additional assumptions, and becomes inaccurate towards the non-linear response range of the sensor, a fully quantitative approach has been criticised. In this work we use the FRET values as direct representation of the in vivo dynamics and relative differences (*Wagner et al., 2015b*). When conversion to absolute concentrations is desired, they should be regarded as an estimate. Here about 2 mM MgATP$^{2-}$ can be estimated for the cytosol at steady state in the dark (*Figures 2* and *5*). This concentration value complements ATP determinations normalised to chlorophyll contents by protoplast fractionation (*Stitt et al., 1982*; *Gardeström and Wigge, 1988*) and by NMR (*Gout et al., 2014*).

## Towards an integrated understanding of energy physiology in development and under stress

CLSM resolved organ, tissue and cell differences to give a detailed MgATP$^{2-}$ map of the Arabidopsis seedling (*Figures 6* and *7*). Similar maps have been generated for hormone distribution (e.g. auxin and abscisic acid *Brunoud et al., 2012*; *Jones et al., 2014*) to study plant development. For central metabolites high-resolution transcriptional analysis has aimed to generate tissue maps (*Chaudhuri et al., 2008*), but direct in vivo mapping across tissues and organs has not been undertaken. Differences in the MgATP$^{2-}$ concentration between tissues and cells have not previously been distinguishable with any reasonable resolution, and the MgATP$^{2-}$ map therefore unveils a surprisingly heterogeneous and dynamic picture. Green tissues showed overall high MgATP$^{2-}$ concentration, which are unlikely due to photosynthetic activity because the seedlings were dark-adapted. Etiolation not only decreased MgATP$^{2-}$ in the green tissues, but also increased levels in the roots. The biological significance of tissue-specific MgATP$^{2-}$ heterogeneity deserves in-depth dissection in the context of metabolism and development in the future. Since the map represents MgATP$^{2-}$, as the main bioavailable form of ATP, differences may, in principle, be due to ATP pool size, pH and/or Mg$^{2+}$. Many proteins bind MgATP$^{2-}$, but the extent to which this affects the buffering of the pool and the free concentration, which is available for binding by the sensor, is currently unknown.

With the appropriate caution, the map provides insight into MgATP$^{2-}$ distribution at system level and the concept sets a reference point for expansion into several dimensions, including time, subcellular compartment, and other genetic, biochemical or physiological parameters. Time-lapse imaging can resolve the dynamic changes in a tissue, both during developmental processes and in response to external stimuli. Our observation that the MgATP$^{2-}$ concentration increases with decreasing root hair growth exemplifies the occurrence of such dynamics and provides evidence for the complex relationship between cellular energy status and growth. Analogous maps can be generated for other subcellular compartments, such as the plastids, to obtain a tissue-resolved map of subcellular MgATP$^{2-}$ heterogeneity. Other sensors, e.g. for pH, Ca$^{2+}$ or glutathione redox potential, may be used and superimposed to gradually build up a comprehensive representation of the cell physiological status of the whole plant. Superimposition with transcriptomic and proteomic maps (e.g., *Chaudhuri et al., 2008*; *Li et al., 2016*) may even allow correlation across the organisational levels of the cell. Such systemic multi-dimension in vivo mapping will provide a novel foundation to modelling attempts of whole plant metabolism and to the understanding of how metabolism underpins plant development.

Disruption of fresh oxygen supply to Arabidopsis seedlings resulted in a gradual decrease in the cytosolic MgATP$^{2-}$ concentration (*Figure 6*), following expectations from previous work on plant extracts and using NMR (*Xia and Saglio, 1992*; *Geigenberger et al., 2000*; *Gout et al., 2001*; *van Dongen et al., 2003*). The decrease occurred in four phases and was fully reversible by re-oxygenation. FRET ratios initially declined slowly, however, after extended block of oxygen supply, a sharp decrease in the MgATP$^{2-}$ concentration occurred to a new plateau level, which may represent the lower sensitivity limit of the probe. The exact kinetics depended on the experimental setup and

the type of tissue, yet the overall response was reproducible (*Figure 10*; *Figure 10—figure supplement 1*). The rapid response and the gradual decrease together support the rationales that the cytosolic $MgATP^{2-}$ pool strictly depends on oxidative phosphorylation, that AK-based buffering may at most delay $MgATP^{2-}$ depletion, but that a sharp $MgATP^{2-}$ depletion can be avoided, probably by restructuring of metabolic fluxes (*Geigenberger, 2003*; *Zabalza et al., 2009*). The experimental setup did not generate a sudden decrease in oxygen availability, but relied on dark respiration for gradual oxygen depletion. The observed responses therefore represent the integrated effects of dynamic progression of hypoxia and the compensating metabolic changes; as they may occur in a situation of sudden (deep) water logging. Hypoxia-associated intracellular acidification may also impact on the sensor response by influencing $MgATP^{2-}$ dissociation. Yet, these pH-induced changes are not artefacts and carry physiological meaning, since a decrease in the concentration of $MgATP^{2-}$ does not only affect the sensor, but also endogenous $MgATP^{2-}$-dependent proteins. Furthermore, the pH controlled $MgATP^{2-}$ depletion treatments indicate only a minor contribution of pH (*Figure 5—figure supplement 3*) and prior work has suggested that cytosolic pH hardly decreases below 7, even under anoxia (*Gout et al., 2001*; *Schulte et al., 2006*). As such, the hypoxia assays demonstrate that ATeam1.03-nD/nA allows continuous monitoring of subcellular $MgATP^{2-}$ pools in response to stress. Since the exact ATP kinetics are shaped by the cellular stress response machinery, they will provide a sensitive and integrated readout for defects or modifications in mutants.

## Conclusions and outlook

We have investigated mitochondrial bioenergetics, plant hypoxia responses and $MgATP^{2-}$ content in plant tissues depending on growth conditions and development. These studies exemplify the versatility of fluorescent ATP sensing to open new doors in plant biology. Their systematic follow up and extrapolation will be required for a systems view of ATP in the future. Although ATP biochemistry has been extensively studied in the last century, surprisingly large gaps remain in our understanding of its dynamics within cells and whole plants. Sensing $MgATP^{2-}$ dynamically and with (sub)cellular resolution adds novel depth to the study of plant metabolism, development, signalling and stress responses. Our understanding of plant microbe interactions, where the biochemistry underpinning the localised and dynamic responses have been notoriously hard to capture, may benefit in particular. Other potential applications include the identification and in vivo validation of ATP transport systems, a better understanding of the coordination between plastids and mitochondria in ATP production, an appraisal of the impact of uncoupling systems, the visualisation of energy parasitism in diatoms, the extension of in situ enzyme assays and the mapping of $MgATP^{2-}$ to other sensor outputs and oxygen gradients in tissues. Optimisation of high affinity ATP sensors for the apoplast could support investigations on extracellular ATP signalling. Additional sensors for AMP, ADP and ATP:ADP ratio can be integrated into the methodological framework introduced here. Each time that a biosensor for a new facet of cell physiology, e.g. for free $Ca^{2+}$ or glutathione redox potential (*Allen et al., 1999*; *Pei et al., 2000*; *Meyer et al., 2007*; *Schwarzländer et al., 2008*; *Krebs et al., 2012*; *Loro et al., 2012*; *Wagner et al., 2015b*), has been introduced into plant research, this has yielded a burst of discovery. We expect ATP sensing to be no exception.

## Materials and methods

### Cloning of sensor constructs and generation of plant lines

The ATeam1.03-nD/nA sequence was PCR-amplified from pENTR1A:ATeam1.03-nD/nA. The leader sequence from *Nicotiana plumbaginifolia* ß-ATPase (*Logan and Leaver, 2000*) for mitochondrial import was fused to the N-terminus by extension PCR. For constitutive plant expression under a CaMV 35*S* promoter, this fusion and the unfused sequence for cyto-nuclear targeting were subcloned into pDONR207 (Invitrogen Ltd, Carlsbad, CA) and ultimately pB7WG2 and pH2GW7, respectively (*Karimi et al., 2002*). For targeting to the plastid stroma, the leader sequence from *Nicotiana tabacum* transketolase (*Wirtz and Hell, 2003*; *Schwarzländer et al., 2008*) and the ATeam1.03-nD/nA sequence were subcloned into pENTR/D-TOPO (Invitrogen Ltd) via NdeI/PstI and BamHI/XbaI restriction sites, respectively. The fusion was cloned into pEarleyGate100 (*Earley et al., 2006*) for 35*S*-driven expression. Primer sequences are detailed in *Supplementary file 1*. The non-fused sequence was also inserted into pETG10A for protein expression *in Escherichia coli* cells.

Agrobacterium-mediated transformation of *Arabidopsis thaliana* (L.) Heynh. (accession Columbia, Col-0) was performed by floral dip (*Clough and Bent, 1998*). Transformants and homozygous lines were selected by chemical resistance and fluorescence intensity. Generation of the NES-YC3.6 Cameleon line was described previously (*Krebs et al., 2012*).

## Chemicals

Chemicals were purchased from Sigma-Aldrich (Taufkirchen, Germany). Stock solutions of ATP, ADP and AMP were freshly supplemented with equimolar concentrations of $MgCl_2$ except for $Mg^{2+}$ titration. All stock solutions were adjusted to assay pH prior to use.

## Plant culture

Where not indicated otherwise, Arabidopsis seedlings were grown from surface-sterilised seeds on vertical plates containing half-strength Murashige and Skoog (MS) medium (*Murashige and Skoog, 1962*) with 1% (w/v) sucrose and 0.8% (w/v) Phytagel under long-day conditions (16 hr 80–120 μmol photons $m^{-2}$ $s^{-1}$ at 22°C, 8 hr dark at 18°C) after stratification at 4°C in the dark. For line preparation of leaf discs, plants were germinated and grown on soil under long-day conditions (17°C, 16 hr 50–75 μmol photons $m^{-2}$ $s^{-1}$, 8 hr dark) after stratification at 4°C in the dark.

## Plant phenotyping

Seeds were stratified at 4°C in the dark for 2 d on half-strength MS medium +1% (w/v) sucrose +1% (w/v) MES +1% (w/v) Phytagel, pH 5.8 and seedlings were grown on the plates vertically side by side with their corresponding controls for 5 d under long-day conditions (16 hr at 22°C and 75–100 μmol photons $m^{-2}$ $s^{-1}$, 8 hr at 18°C and darkness). Primary root length was documented and quantified using ImageJ. Plants were then individually transferred to Jiffy-pots, randomly distributed on standard greenhouse flats and grown in long-day (16 hr at 19°C and 60–80 μE $m^{-2}$ $s^{-1}$, 8 hr at 17°C and darkness) growth chambers. Leaf rosette development was documented photographically and rosette size was analysed with the custom Leaf Lab tool (Version 1.41) as described previously (*Wagner et al., 2015b*). Height of the primary inflorescence was systematically captured with a camera and quantified using ImageJ. Siliques were manually counted when the first siliques turned yellow but had not yet shattered.

## Purification of ATeam1.03-nD/nA

*E. coli* strain Rosetta 2 (DE3) carrying pETG10A-ATeam1.03-nD/nA was grown in lysogeny broth (LB; *Bertani, 1951*) medium at 37°C to an $OD_{600}$ of 0.2. The culture was transferred to 20°C until cells reached an $OD_{600}$ of 0.6–0.8. Expression of 6×His-ATeam1.03-nD/nA was induced by isopropyl β-D-1-thiogalactopyranoside at a final concentration of 0.2 mM overnight at 20°C. Cells were collected by centrifugation at 4000 *g* for 10 min at 4°C and the pellet was resuspended in lysis buffer (100 mM Tris-HCl, pH 8.0, 200 mM NaCl, 10 mM imidazole) supplemented with 1 mg/mL lysozyme, 0.1 mg/mL DNaseI (Roche, Mannheim, Germany) and cOmplete protease inhibitor cocktail (Roche). After 30-min incubation on ice, cells were sonicated (3 × 2 min, 40% power output, 50% duty cycle). The lysate was centrifuged at 40,000 *g* for 40 min at 4°C and the supernatant was loaded onto a Ni-NTA HisTrap column (GE Healthcare, Freiburg, Germany) and proteins were eluted with an imidazole gradient (10–200 mM in 100 mM Tris-HCl, pH 8.0, 200 mM NaCl) using an ÄKTA Prime Plus chromatography system (GE Healthcare). Fractions containing ATeam1.03-nD/nA were pooled, concentrated by ultrafiltration and applied to a HiLoad 16/600 Superdex 200 column (GE Healthcare) pre-equilibrated with 20 mM Tris-HCl, pH 8.0, 150 mM NaCl. Fractions containing intact ATeam1.03-nD/nA were pooled, concentrated by ultrafiltration, supplemented with 20% (v/v) glycerol and stored at −86°C.

## Characterisation of ATeam1.03-nD/nA in vitro

Concentration of purified ATeam1.03-nD/nA was quantified according to *Bradford (1976)*. Protein at a final concentration of 1 μM was mixed with basic incubation medium (0.3 M sucrose, 5 mM $KH_2PO_4$, 50 mM TES-KOH, pH 7.5, 10 mM NaCl, 2 mM $MgSO_4$, 0.1% (w/v) BSA) for all in vitro assays, except for the $Mg^{2+}$ titrations, in which $MgSO_4$ was omitted from the basic incubation medium. A FP-8300 spectrofluorometer (Jasco, Gross-Umstadt, Germany) at 25°C was used to excite

mseCFP at 435 ± 5 nm and emission spectra between 450 to 600 nm were recorded with a bandwidth of 5 nm. Venus/CFP ratios were calculated as measure of FRET efficiency from the fluorescence emission intensities at 527 nm (cp173-Venus) and 475 (mseCFP) nm after blank correction.

## Mitochondrial isolation and oxygen consumption assays

Arabidopsis seedlings were grown in hydroponic pots for 14 days as described by *Sweetlove et al. (2007)* under long-day conditions (16 hr 50–75 μmol photons m$^{-2}$s$^{-1}$ at 22°C; 8 hr dark at 18°C). Seedling mitochondria were isolated as described by *Sweetlove et al. (2007)* and *Schwarzländer et al. (2011)*. Oxygen consumption was assayed as described by *Sweetlove et al. (2002)* and *Schwarzländer et al. (2009)* using two Clark-type electrodes (Oxytherm, Hansatech, Norfolk, UK).

## Multiwell plate reader-based fluorimetry

ATeam1.03-nD/nA was excited with monochromatic light at a wavelength of 435 ± 10 nm in a CLAR-IOstar plate reader (BMG Labtech, Ortenberg, Germany). Emission was recorded at 483 ± 9 nm (mseCFP) and 539 ± 6.5 nm (cp173-mVenus) using transparent 96-well plates (Sarstedt, Nümbrecht, Germany). The internal temperature was kept at 25°C, and the plate was orbitally shaken at 400 rpm for 10 s after each cycle. For assays with isolated mitochondria, 20 μg total protein in basic incubation medium (0.3 M sucrose, 5 mM KH$_2$PO$_4$, 50 mM TES-KOH, pH 7.5, 10 mM NaCl, 2 mM MgSO$_4$, 0.1% (w/v) BSA) were supplemented with 1 μM purified ATeam1.03-nD/nA in a total volume of 200 μL per well. Fluorescent background of the basic incubation medium was recorded and subtracted from all data before analysis. For in vivo experiments with Arabidopsis seedlings and leaf discs, plant material was submerged in 10 mM MES, pH 5.8, 5 mM KCl, 10 mM MgCl$_2$, 10 mM CaCl$_2$. Plates were kept in the dark for at least 30 min before recording to minimise potential effects of active photosynthesis. TES buffer was replaced with Bis-Tris-HCl (covering pH values between 6.0 and 7.0) or Tris-HCl (covering pH values between 7.5 and 8.5) where indicated. To restrict supply with oxygen, plates were sealed with ultra-clear films for qPCR (VWR, Langenfeld, Germany).

## Confocal laser scanning microscopy and analysis

Confocal imaging was performed at 20°C using a Zeiss LSM780 microscope and a ×5 (EC Plan-Neofluar, 0.16 N.A.),×10 (Plan-Apochromat, 0.3 N.A) or ×40 lens (C-Apochromat, 1.20 N.A., water immersion) using the procedure described previously (*Wagner et al., 2015a*). ATeam1.03-nD/nA was excited at 458 nm and fluorescence of mseCFP and cp173-mVenus were measured at 465–500 nm and 526–561 nm, respectively, with the pinhole set to three airy units. Chlorophyll fluorescence was collected at 650–695 nm. MitoTracker Orange was excited at 543 nm and emission was recorded at 570–619 nm. Plants were dark-adapted for at least 30 min before image acquisition. Single plane images were processed with a custom MATLAB-based software (*Fricker, 2016*) using *x,y* noise filtering and fluorescence background subtraction.

## Light sheet fluorescence microscopy and analysis

Hair cell growth on roots from six-day-old seedlings was followed for 10 min with images acquired every 3 s. Each image was generated as a maximum intensity projection (MIP) of 15 stacks spaced by 3 μm and the MIP acquisition time was 1 s, based on 50 ms exposure per stack and data transfer time. The LSFM used was specifically designed to study plant roots (*Costa et al., 2013*; *Candeo et al., 2017*). The source was a single-mode fibre-coupled laser emitting radiation at 442 nm (MDL-III-442, CNI) to excite ATeam mseCFP. A cylindrical lens in combination with a 10× water-dipping microscope objective (Olympus UMPLFLN 10×W, 0.3 N.A.) created a thin sheet of laser light on the sample (5-μm-thick and 800-μm-high in the vertical direction). The detection unit consisted of a 20× water-dipping microscope objective (Olympus UMPLFLN 20×W, 0.5 N.A.), held orthogonally to the excitation axis. An image splitter (dichroic filter at 505 nm, DMLP505, Thorlabs, Newton, NY; band-pass filters, MF479-40 and MF535-22, Thorlabs, Newton, NY) and a fast camera (Neo 5.5 sCMOS, ANDOR, Belfast, UK) enabled the simultaneous wide-field acquisition of two spectrally different images, as required for a FRET indicator. The system provided an intrinsic optical sectioning with minimal light exposure of the sample, permitting a fast volumetric acquisition and long-term imaging over a wide-field of view with single-cell detail. The seedlings were grown vertically in

fluorinated ethylene propylene tubes filled with half-strength MS medium, 0.1% (w/v) sucrose; 0.05% (w/v) MES, pH 5.8 (Tris), solidified by 0.5% (w/v) Phytagel as described in *Candeo et al. (2017)* and kept in an imaging chamber filled with liquid MS-based medium lacking sucrose (half-strength MS medium; 0.05% (w/v) MES; pH 5.8 (Tris)).

The Venus/CFP ratio at the tip of single root hairs was extracted with FIJI (https://fiji.sc/) after image registration by using the Fiji plugin 'Template Matching' (https://sites.google.com/site/qing-zongtseng/template-matching-ij-plugin). Finally, the Matlab Fast Fourier Transform was used to calculate the power spectral densities from the normalised Venus/CFP time courses (*Candeo et al., 2017*). To quantify the average speed growth of root hairs we measured their elongation during the entire acquisition and divided by the time (10 min).

## Acknowledgements

We thank Takeharu Nagai (Osaka, Japan) for the kind gift of the pENTR1A:ATeam1.03-nD/nA construct, Jan Riemer (Cologne, Germany), Elisa Petrussa and Valentino Casolo (Udine, Italy) for important discussions, and Silvina Paola Denita Juárez (Buenos Aires, Argentina) and Andrea Magni (Milan, Italy) for assistance with the LSFM experiments.

## Additional information

### Funding

| Funder | Grant reference number | Author |
|---|---|---|
| Deutsche Forschungsgemeinschaft | SCHW1719/1-1 | Markus Schwarzländer |
| Deutsche Forschungsgemeinschaft | GR4251/1-1 | Christopher Grefen |
| Deutsche Forschungsgemeinschaft | GRK 2064 | Andreas J Meyer Markus Schwarzländer |
| Deutsche Forschungsgemeinschaft | ME1567/9-1, SPP1710 | Andreas J Meyer |
| Deutsche Forschungsgemeinschaft | SCHW1719/5-1 | Markus Schwarzländer |
| Bioeconomy Science Center, NRW | CoSens | Andreas J Meyer Markus Schwarzländer |
| Ministero dell'Istruzione, dell'Università e della Ricerca | RBFR10S1LJ_001 | Alex Costa |
| Piano di Sviluppo di Ateneo | | Alex Costa |
| Deutscher Akademischer Austauschdienst | | Thomas Nietzel Stephan Wagner |
| European Social Fund | Operational Programme 2007/2013 | Valentina De Col |
| European Commission | Erasmus+ | Valentina De Col |
| Human Frontier Science Program | RPG0053/2012 | Mark D Fricker |
| Leverhulme Trust | RPG-2015-437 | Mark D Fricker |
| Independent Research Fund Denmark - Natural Sciences | | Ian Max Møller |
| Innovation and Technology Commission | Partner State Key Laboratory | Chia Pao Voon |
| Ministero dell'Istruzione, dell'Università e della Ricerca | PRIN2010CSJX4F | Marco Zancani |
| European Commission | Laserlab-Europe EU-H2020 654148 | Andrea Bassi |

The funders had no role in study design, data collection and interpretation, or the decision to submit the work for publication.

## Author contributions

VDC, Formal analysis, Investigation, Methodology, Writing—original draft; PF, Formal analysis, Investigation, Methodology, Writing—review and editing; TN, ME, CPV, IS, Investigation, Methodology; ACa, Formal analysis, Investigation, Methodology; MDF, Software, Funding acquisition, Writing—original draft, Writing—review and editing; CG, Funding acquisition, Methodology, Writing—original draft, Writing—review and editing; IMM, Funding acquisition, Validation, Writing—original draft, Writing—review and editing; AB, Resources, Supervision, Funding acquisition, Investigation, Methodology; BLL, Conceptualization, Supervision, Funding acquisition, Writing—review and editing; MZ, Supervision, Funding acquisition, Writing—review and editing; AJM, Resources, Supervision, Funding acquisition, Writing—review and editing; ACo, Conceptualization, Funding acquisition, Investigation, Methodology, Writing—original draft, Writing—review and editing; SW, Conceptualization, Formal analysis, Supervision, Investigation, Visualization, Methodology, Writing—original draft, Project administration, Writing—review and editing; MS, Conceptualization, Resources, Formal analysis, Supervision, Funding acquisition, Validation, Investigation, Methodology, Writing—original draft, Project administration, Writing—review and editing

## Author ORCIDs

Valentina De Col, http://orcid.org/0000-0003-0895-969X
Philippe Fuchs, http://orcid.org/0000-0001-6379-853X
Thomas Nietzel, http://orcid.org/0000-0002-1934-1732
Chia Pao Voon, http://orcid.org/0000-0002-4959-2559
Alessia Candeo, http://orcid.org/0000-0001-9597-3056
Mark D Fricker, http://orcid.org/0000-0002-8942-6897
Christopher Grefen, http://orcid.org/0000-0002-5820-4466
Andrea Bassi, http://orcid.org/0000-0002-5017-0775
Boon Leong Lim, http://orcid.org/0000-0002-2720-2353
Marco Zancani, http://orcid.org/0000-0003-4354-748X
Andreas J Meyer, http://orcid.org/0000-0001-8144-4364
Alex Costa, http://orcid.org/0000-0002-2628-1176
Stephan Wagner, http://orcid.org/0000-0001-5369-7911
Markus Schwarzländer, http://orcid.org/0000-0003-0796-8308

# Additional files

## Supplementary files

• Supplementary file 1. Oligonucleotides for generation of expression constructs

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
