## [Decision Letter]

Thank you for submitting your article "ATP sensing in living plant cells reveals tissue gradients and stress dynamics of energy physiology" for consideration by *eLife*. Your article has been reviewed by two peer reviewers, and the evaluation has been overseen by Christian Hardtke as the Senior and Reviewing Editor. The following individuals involved in review of your submission have agreed to reveal their identity: Teva Vernoux (Reviewer #1) and Alisdair Fernie (Reviewer #2).

In your revision, please pay particular attention to complete documentation of the phenotypes and adequate, more complete control panels. Specifically, please provide more comprehensive evidence that there is no phenotype in the cytosolic/chloroplastic lines. Also, please carefully review a potential effect of the liquid culture, which might by itself create normoxic conditions. Finally, we would like to ask you to address the reviewers' comments, which you find below, point by point.

*Reviewer #1:*

This manuscript describes the transfer and characterisation in plants (*Arabidopsis*) of a FRET sensor for ATP (more precisely ATP-Mg2-). The sensor is further used to analyse mitochondrial ATP production and to map ATP in plantelets. The authors also use it to analyse changes in ATP levels during anoxia, showing that it can be used to follow the energetic response during stresses.

The work reported is of high quality and the data presented are very convincing. This ATP FRET sensor will surely be widely used to understand ATP biochemistry at high spatio-temporal definition and how it is integrated in planta.

I have only relatively minor comments (below, starting with the most important ones) that I think the authors should be able to address easily and that will strengthen the manuscript:

– One (real) concern with the manuscript is the fact that the number of transgenics is very low (3 / sub cellular localisation). Considering this, the authors need more than just an image of a single rosette (with no control shown!) to say that the development is not affected in the cytosolic and chloroplastic lines. This is all the more important that there is a very strong phenotype for the mitochondrial lines. I suggest the authors to quantify for all the lines they have various parameters of root development in vitro and of shoot development for plants on soil (size of rosettes, height of inflorescence, number of siliques). If they can add extra transgenic lines it would obviously strengthen the data.

– The authors should make clear already in the Abstract that it is a sensor for ATP-Mg2- rather to highlight its specificity and indicate why being able to detect ATP-Mg2- is pertinent. They should also recall this point in the discussion a bit more thoroughly.

– For the analysis of the effects of hypoxia, I am a bit concerned by the fact that the assay uses plants that are grown in liquid. This is in itself should induce hypoxia if I am not mistaken. Then the results observed would correspond to increasing the level of stress rather than no stress/stress comparison. Could the authors compare the distribution of the sensor for plants grown on plates to the one in liquid to estimate the effect of the liquid culture? Minimally (and if I am not wrong) they should point out this limit of their analysis.

– The authors claim an inverse correlation between ATP and growth in root hair with a support from Figure 8. The R square on the figure is 0.13. Nothing can be concluded form such a low R square that suggests that the correlation is not better than what would be obtained in a random situation. Is there a mistake with this value? If it is the right one the authors should probably be more careful and conclude that what they say is true for the highest and lowest growth values.

*Reviewer #2:*

The manuscript by De Col et al., presents a validated plant ATP sensor. It additionally maps ATP tissue gradients and subcellular distribution using cytosolic, mitochondrial and chloroplastic ATP sensors providing perspective of their use within the text. The mitochondrial plants are dwarven which may confound their utility in some applications, to get around this they demonstrate an ex situ assay for isolated mitochondria. These studies thus all demonstrate the fact that these sensors can be used within live plants. In addition the authors also follow the kinetics of variation in ATP levels following the progression of hypoxia. In short this paper is a real tour-de-force not only does it demonstrate the usefulness of this technology but it already put is to work to address some of the most important questions around concerning ATP. The sheer amount of work included here is laudable as are the number of careful controls I went through the fine details of each experiment but have no disagreements or uncertainties with them as presented. I also have no request for further experimentation but do list a few of the additional areas into which this technology could be developed, applied and contrasted below should the authors feel that they are worth adding to their perspective section. That said I enthusiastically echo their suggestion that this technology will likely assist in our understanding of what is unarguably one of the most important cellular metabolites of plants and indeed all living systems.

---

## [Author Response]

*Reviewer #1:*

*This manuscript describes the transfer and characterisation in plants (Arabidopsis) of a FRET sensor for ATP (more precisely ATP-Mg2-). The sensor is further used to analyse mitochondrial ATP production and to map ATP in plantelets. The authors also use it to analyze changes in ATP levels during anoxia, showing that it can be used to follow the energetic response during stresses.*

*The work reported is of high quality and the data presented are very convincing. This ATP FRET sensor will surely be widely used to understand ATP biochemistry at high spatio-temporal definition and how it is integrated in planta.*

*I have only relatively minor comments (below, starting with the most important ones) that I think the authors should be able to address easily and that will strengthen the manuscript:*

*– One (real) concern with the manuscript is the fact that the number of transgenics is very low (3 / sub cellular localisation). Considering this, the authors need more than just an image of a single rosette (with no control shown!) to say that the development is not affected in the cytosolic and chloroplastic lines. This is all the more important that there is a very strong phenotype for the mitochondrial lines. I suggest the authors to quantify for all the lines they have various parameters of root development* in vitro *and of shoot development for plants on soil (size of rosettes, height of inflorescence, number of siliques). If they can add extra transgenic lines it would obviously strengthen the data.*

We are grateful for this suggestion and have performed detailed phenotypic characterisation of the homozygous sensor lines presented in the first submission (i.e., cytosolic lines #1.1 and #3.6, plastidic lines #1.1 and #2.1 and mitochondrial lines #1.1 and #4.8) covering all suggested parameters. We determined root length of vertically grown plants side-by-side with wild type controls, and quantified rosette sizes of the plants after transfer to soil until the majority of individuals showed open flowers. In the reproductive phase, we determined the inflorescence height and number of siliques at individual time points. We additionally included a comparison of these parameters between fluorescent (+/+ and +/-) and non-fluorescent (-/-) individuals from two heterozygous lines expressing ATeam in the cytosol and one heterozygous line expressing ATeam in the plastid. As mitochondrial localisation of the sensor consistently confirmed the strong developmental phenotype, we decided against the characterisation of an additional line and do not generally recommend these plants for physiological measurements. We have added the phenotyping data to Figure 1 as Figure supplements 2 and 3, and cover them in the Results and Materials and methods section

*– The authors should make clear already in the Abstract that it is a sensor for ATP-Mg2- rather to highlight its specificity and indicate why being able to detect ATP-Mg2- is pertinent. They should also recall this point in the discussion a bit more thoroughly.*

We have adjusted the text accordingly and now refer more consistently and specifically to MgATP^2-^ throughout the text, to emphasise the chemical specificity of the sensor as well as the physiological significance of MgATP^2-^.

*– For the analysis of the effects of hypoxia, I am a bit concerned by the fact that the assay uses plants that are grown in liquid. This is in itself should induce hypoxia if I am not mistaken. Then the results observed would correspond to increasing the level of stress rather than no stress/stress comparison. Could the authors compare the distribution of the sensor for plants grown on plates to the one in liquid to estimate the effect of the liquid culture? Minimally (and if I am not wrong) they should point out this limit of their analysis.*

We have repeated the experiment with seedlings grown vertically on half-strength MS medium solidified with agar. The ATP dynamics under oxygen shortage appeared generally conserved. The exact timing of events differed slightly between experiments with oxygen-limited seedlings reaching minimal ATP levels more slowly when grown on plate than in hydroponic culture. This might indeed indicate that hydroponically grown plants are affected by mild hypoxia already before the experiment. Differences could additionally result from more general effects of the growth method such as individual biomass or the shoot-to-root ratio. A related and more general weakness of the technical setup is that the plants are submerged during the experiment. Hence, each experiment includes a control population of submerged plants that were still supplied with oxygen (not sealed). We have replaced the data from the hydroponically grown seedlings with the new data from plate-grown seedlings in Figure 10, but still show the other experiment as Figure 10—figure supplement 1 to highlight the robustness of the assay and the response pattern.

*– The authors claim an inverse correlation between ATP and growth in root hair with a support from Figure 8. The R square on the figure is 0.13. Nothing can be concluded form such a low R square that suggests that the correlation is not better than what would be obtained in a random situation. Is there a mistake with this value? If it is the right one the authors should probably be more careful and conclude that what they say is true for the highest and lowest growth values.*

We are grateful for the remark and agree that the low coefficient of determination (R²) carries insufficient meaning. We have therefore included an F-test as appropriate statistical assessment of the linear regression, which suggest that the general correlation between root hair growth rate and the ATeam Venus/CFP ratio was significant at the 0.05 level (p-value of 0.011). Together this suggests that the variance in root hair growth cannot largely be explained by the ATP levels and likely depends on a number of variables, but the correlation between growth and ATP levels is statistically valid, not only for the hair cells showing fastest and slowest growth, but also for the full dataset. We have revised Figure 8 and checked the relevant text passages to make sure that our interpretation is sufficiently cautious.

*Reviewer #2 (Major comments (Required)):*

*The manuscript by De Col et al., presents a validated plant ATP sensor. It additionally maps ATP tissue gradients and subcellular distribution using cytosolic, mitochondrial and chloroplastic ATP sensors providing perspective of their use within the text. The mitochondrial plants are dwarven which may confound their utility in some applications, to get around this they demonstrate an ex situ assay for isolated mitochondria. These studies thus all demonstrate the fact that these sensors can be used within live plants. In addition the authors also follow the kinetics of variation in ATP levels following the progression of hypoxia. In short this paper is a real tour-de-force not only does it demonstrate the usefulness of this technology but it already put is to work to address some of the most important questions around concerning ATP. The sheer amount of work included here is laudable as are the number of careful controls I went through the fine details of each experiment but have no disagreements or uncertainties with them as presented. I also have no request for further experimentation but do list a few of the additional areas into which this technology could be developed, applied and contrasted below should the authors feel that they are worth adding to their perspective section. That said I enthusiastically echo their suggestion that this technology will likely assist in our understanding of what is unarguably one of the most important cellular metabolites of plants and indeed all living systems.*

We are grateful for this positive assessment and have incorporated several of the listed perspective aspects.

We would like to thank both reviewers for their clear, helpful and rigorous comments, which have helped us to increase the quality and accessibility of the submission. The fact that both reviewers share our enthusiasm about potential future applications fills us with optimism that our work will be actively picked up on by the research community.